# High microbial diversity in glacial habitats uncoupled from the specialized microbiomes of resident chironomid fauna

Valeria Lencioni[1], Alessandra Tondello[2], Isabel Martinez-Sañudo[2], Saptarathi Deb[2], Lucia Giagnoni[2], Augusto Zanella[3], Giuseppe Concheri[2], Piergiorgio Stevanato[2], Luca Mazzon[2], Andrea Squartini[2]*

1 Climate and Ecology Unit, Research and Museum Collection Office, MUSE-Museo delle Scienze, Trento, Italy, 2 Department of Agronomy, Animals, Food, Natural Resources, and Environment (DAFNAE), University of Padova, Viale dell'Università 16, Legnaro, Padova, Italy, 3 Department Land Environment Agriculture and Forestry, University of Padua, Viale dell'Università 16, Legnaro, Italy

* squart@unipd.it

## Abstract

Ecosystems associated with retreating glaciers are undergoing rapid transformation in the context of a changing climate. Invertebrate fauna, including the Chironomidae insect family (non-biting midges), is part of the active biology that characterizes glaciers and their surrounding habitats. The underlying microbiology, present in both the insects' guts and their physical habitat, represents a critical interface, controlling, on the one hand, the basis of nutrient geochemical cycling and, on the other, the health and nutritional physiology of its hosts. We aimed to assess the extent to which insect-borne bacteria resemble those found in icemelt water and the surrounding wet and terrestrial environments, in order to determine also whether the bacteria found associated with the insects could be interpreted mainly as specific dwellers, putatively involved with active physiological functions, or also as transient cells taken in for other purposes. To this end, we analyzed physical and biological samples from ten different chironomid species in two glacier-fed streams, one proglacial pond, and the surrounding habitats of three glacier systems (Agola, Amola and Mandrone) in the Italian Alps. The samples were analyzed using culture-independent amplified 16S rRNA gene bacterial metabarcoding sequencing. The bacterial diversity in glacial habitats was unexpectedly high, with numbers of sequence variants similar to those recorded in temperate, lowland, productive soils, and almost seven times higher than those found in insects. There was minimal coincidence in sequence variants between insects and habitats, amounting to just 4.9% shared cases, and the few taxa found in both insects and habitats were mostly overrepresented in the former. Additionally, there were no significant differences between insect species or between insect communities from different sites. A number of taxa occurring uniquely in insects or habitats showed peculiarities at all taxonomic levels, including specific phyla. Evidence

**Data availability statement:** The sequence data were deposited in the NCBI Sequence Reads Archive (SRA) under code PRJNA1222183.

**Funding:** Grant BioMiti 2018, issued from Adamello-Brenta Natural Park, Italy, awarded to A.Z. The funders had no role in study design, data collection and analysis, decision to publish, or preparation of the manuscript.

**Competing interests:** The authors have declared that no competing interests exist.

of microevolutionary distinctness was observed in the form of sequence variants assigned to the same taxonomic name that differed in specific sequence bases and were found to be partitioned either in insect or environmental samples. Interestingly, sequence variants found in both insects and environments scored higher in bioinformatic identification, reaching deeper assigned ranks compared to variants occurring only in insects or the environment. In essence, the compared insect and environmental communities showed a very low level of symmetry and consequently a very high level of specificity to one or the other condition. The data provided limited support about the diet of chironomids in relation to the microbiota of the habitat. This suggests that the food resource for these insects likely consists mostly of dissolved organic matter and detritus of various origins, rather than intact bacteria with sequenceable genomes, as is the case elsewhere with ruminant herbivores or birds.

## Introduction

Within the different aquatic insects, the object of this research are the non-biting midges (Diptera Chironomidae), which represent the insect family most vastly distributed in cold freshwaters, that also results often the dominant one in glacier-fed streams [1–4]. Carbon stable isotope studies have shown that feeding patterns in chironomids are influenced by past glacier dynamics; [5]. The food chain length in these habitats is relatively short [6–8] chironomids playing a crucial role as primary consumers [3,4]. Several studies refer to the diet of these insects [9–11] and few on their microbiota [12]. The dietary habits of chironomids are influenced by the type and quality of food present in their environment. For instance, habitats rich in epiphyton or sandy areas tend to support more algae, while organic sediments offer higher levels of detritus. Most chironomid larvae reside at the bottom, living in niches from which they extract nutrients suspended in the water. Some species forage by scraping or gathering organic matter from sediments or rocky surfaces in lakes and rivers. A few are carnivorous, preying on other chironomid larvae [13]. As adults, chironomids consume substances that contain sugars such as sucrose and glucose [14].

Glacial environments host a variety of invertebrate animals, which interact closely with microbial communities. In these animals, microbes are primarily found in the gut and may be either resident, contributing to physiological or symbiotic functions, or transient, originating from ingested food. Since microbiome studies are carried out indirectly by metagenome analyses, species barcodes can also be useful for inferring an animal's diet. However, distinguishing between live-hosted microbiota and consumed food remnants by means of DNA sequencing can be challenging. The insect gut microbiome is, at first, highly dependent on the host's developmental stage [15]. Despite a conserved core, even the often neglected individual variation is a significant source of intraspecific variation [16]. Conversely, host species has been identified as the main driver of the structure and diversity of insect-associated bacterial communities [17]. Hosted microbes play several roles in insects, including digestion of substrates, provision of essential nutrients detoxification, development,

and pathogen resistance. The ranking of these phenotypes assigns a prominent position to the first two physiology-related aspects [18]. While most insects harbour relatively few microbial species compared to the mammals' guts, but some species can host guilds of highly specialized bacteria, while others can be sparsely and opportunistically colonized by stochastic processes involving bacteria common in other environments. Indeed, with the exception of social insects, a clear hindrance to the establishment of stable co-evolutionary associative patterns is the lack of dependable transmission routes to the offspring [19]. Focusing more specifically on the bacterial interactions with freshwater insects, the strongest determinant in shaping the inner microbiota community structure was found to be the feeding functional group variable (predators, omnivores, filtering collectors, gathering collectors, scrapers, shredders/detritivores), leading to clear distinctions also in comparisons within the same stream [20,21].

Rouf and Rigney [22] were the first to identify endogenous bacterial populations in chironomids, specifically in *Chironomus plumosus* larvae sampled from Lake Winnebago, WI, USA producing a list of dozens of endogenous bacteria genera among which *Pseudomonas, Serratia* and *Providencia*. Halpern & Senderovich [12]studied endogenous bacterial communities associated with *Chironomus transvaalensis* egg masses and larvae with a focus on bacteria with a role in protecting its host from toxic metals or other pollutants. Other authors [23] investigated the bacterial community in glacial habitats, even through the microbiota analysis of chironomids, addressing microbial genera richness, indicator genera richness, and *Polaromonas* relative abundance, all of which declined with increasing larval Chironomidae abundance. They highlighted the importance of larval insects in structuring microbial communities in supraglacial pools. Zhang and coworkers [24] evaluated the biodiversity and environmental factors associated with the glacier microbiomes of seven contrasting habitats (water, epilithic biofilm, cryoconite, mat, ice, sediment and permafrost soil) but not in biota. Considering whether one could expect that the chironomids' diet could include a substantial direct intake of bacteria from the habitat, there are premises to cast hypotheses in this direction. Other reports indicated that bacterial composition within aquatic invertebrates varied with taxonomy, habitat, diet, and time of sample collection [25].

To our knowledge no prior papers referred to the microbiota of glacial chironomids, at the insect species level, nor analyzed differences between bacterial communities of different terrestrial and aquatic glacial microhabitats.

Despite the common perception of low temperatures as preserving biological inactivity, glacial environments within an increasingly warmer climate are the forefront of an awakened dynamics for microbes because of the fluxes of nutrient transfer from newly melted ice. The downstream availability of redissolved products from eroded sediments and their fast leaching can foster the biogeochemical elemental cycling. In local glacial and proglacial ecosystems this stirs the involvement of complex microbial consortia [26]. In glacier-fed epilithic biofilms, typical recurring taxa include phyla such as Cyanobacteria, Proteobacteria and Bacteroidetes, and genera, such as *Polaromonas* and *Flavobacterium,* are particularly abundant [27]. In supraglacial habitats, photosynthetic cyanobacteria, as the candidate phylum Patescibacteria and the genus *Ferruginibacter* have also been reported [28]. Glacial streams are found to have higher species richness than their corresponding proglacial lakes, which in turn show higher species diversity [29]. Community assembly processes in proglacial streams were dominated by homogeneous selection, and the effect of physical distance from the glacier on taxa under selection in the glacier-fed sites is limited, presumably due to the strong hydrological connectivity [30]. In the melting glaciers' forefronts microbial diversity is reported to increase along with time since glacier retreat, while beta-diversity between surface and deep layers was found to decrease with time [31]. The time since deglaciation has been identified as the primary driver of the deterministic changes that shape microbial communities [32]. Studies on successional patterns at different taxonomic levels during deglaciation have been meta-analyzed and reviewed [33].

The present study aimed to characterize the bacterial communities associated with 48 samples featuring 10 different species of chironomids collected in three glaciers in the Italian Alps using culture-independent 16S rRNA gene metabarcoding.

The working hypothesis of this investigation was that there could be one of two possible statuses: A) Symmetry (high overlap between communities) i.e., the Chironomids' microbiome mirrors that of the environment, as microbes are

indiscriminately acquired; B) Specificity (high degree of divergence between communities), i.e., chironomids fractionate and select which bacteria would be established in their sequencing-detectable microbiome. We assumed that the prevailing scenario between these possibilities would be revealed when comparing the insect microbiota with its outer environmental counterpart.

To address the above point we undertook this research, whose detailed scopes were: (1) assessing the degree of overlap between the microbiota of the surrounding environment and that of the populations of chironomids living in it; (2) comparing the bacterial assemblages of the different chironomid species with those of the three different sites to verify the possible role of variables as host taxonomy and geographical distance in shaping microbial community structure; (3) inferring information about the insect diet through the identification of potentially conserved patterns of occurrence between the niches of the host gut and the outside nutritional resources..

## Materials and methods

### Study area sites, sampling, and insect identification

The research was carried out in three high mountain sites located in the Italian Alps within the Adamello-Brenta Natural Park territory (Trentino-Alto Adige Region). The permit to sample from the site has been released by Dr. Cristiano Trotter, the Director of Parco Naturale Adamello Brenta, Via Nazionale, 24, 38080 Strembo (TN). Insect larvae were collected in two glacier-fed streams, one fed by the Mandrone glacier (46.15969 N, 10.54022 E, 2686 m a.s.l.), and one by the Amola glacier (46.25743 N, 10.75638 E, 2720 m a.s.l.) and in the Agola proglacial pond facing the Agola glacier front (46.14744 N, 10.87101 E, 2675 m a.s.l.) (Fig 1).

The exact features of each site are shown in Table 1.

The detailed sample list is shown in Table 2.

Chironomid larvae were collected from their water habitats using a hand-held pond net fitted with a 100 μm mesh. Sampling was conducted during daylight hours. The net was gently swept through the water body. After each sweep, the contents of the net were immediately transferred into white plastic trays filled with site water to facilitate visual sorting. Chironomid larvae were manually isolated using fine forceps and placed into sterile 1.5 mL microcentrifuge tubes containing 95% ethanol for preservation. Taxonomic identification was initially done via a morphological approach (microscopic examination of larval features). When morphology was not enough for an appropriate identification, we performed molecular identification of the insect by extracting the larval DNA and amplifying the mitochondrial cytochrome oxidase I (COI) gene, following the protocol of [34]. Briefly, genomic DNA was extracted from the larval body using a QIAGEN DNAeasy Blood & Tissue Kit, following a slightly modified protocol for the purification of total DNA from animal tissue. In particular, 10 μl of a 1 M DTT (dithiothreitol) solution were added to enhance the removal of chitin, after which the samples were incubated overnight at 56 °C for the initial lysis step. The final elution was performed in 50 μl of AE buffer to increase the final DNA concentration. The mitochondrial (COI gene were amplified and sequenced using the primers LCO1490 GGTCAACAAATCATAAAGATATTGG and HCO2198 TAACTTCAGGGTGACCAAAAAATCA.

Adult chironomids were occasionally encountered, and two specimens, collected by tweezers, one on the Mandrone glacier and one on the Amola glacier (Table 2) were included in the analysis. Environmental samples were obtained by collecting material directly into 50 ml sterile Falcon tubes, except the slime on submerged stones, from which the material was removed by scraping with a fine brush and suspended in 20 ml of physiological sterile solution.

In total, the collected material included 36 larvae, 2 adults. and 12 environmental samples.

Total DNA from the habitat samples was obtained using the DNeasy Power Soil Pro Kit (Qiagen GmbH, Hilden, Germany, DE), following the manufacturer indications. The extracted and purified nucleic acids were quantified employing a Qubit Flex fluorometer (Thermo Fisher Scientific, Carlsbad, CA) paired with the Qubit 1x dsDNA High Sensitivity Assay Kit (Thermo Fisher Scientific, Carlsbad, CA).

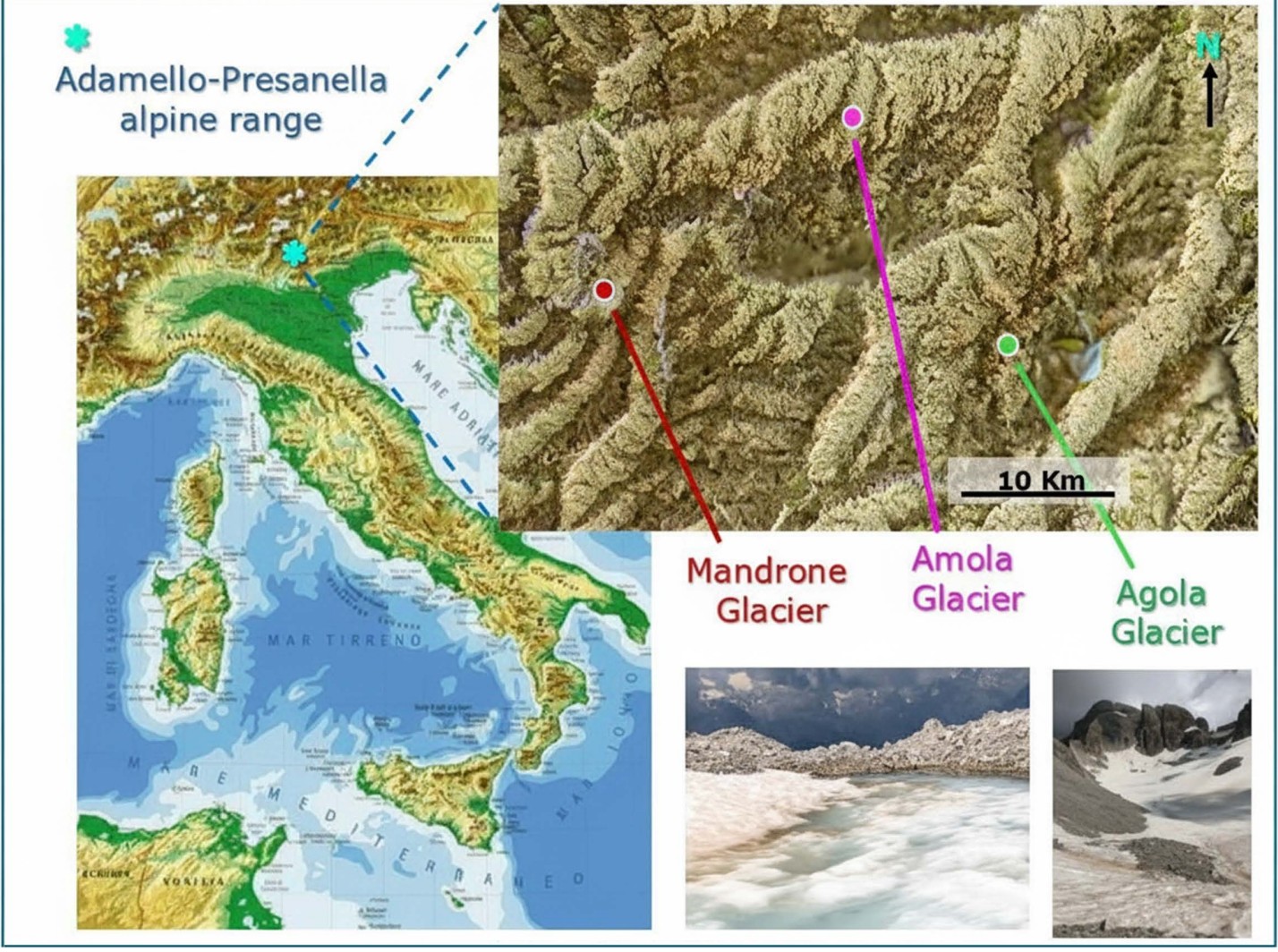

**Fig 1. Location and sampling areas.** The pictures show the map of Italy (left) and that of the area encompassing the three glaciers. The two lower photographs show details of the Agola glacier (right) and its nearby proglacial pond (center).

## Metabarcoding of the bacterial 16S rRNA gene

Library preparation was carried out using a 16S Ion Metagenomics Kit (Thermo Fisher Scientific, Waltham, MA, USA). The Kit is designed for multi-amplicon analyses of polybacterial samples using Ion Torrent™ sequencing technology. It includes two primer sets that selectively amplify seven hypervariable regions of the 16S region in bacteria in two separate reactions. The combination of the two primer pools allows for sequence-based identification of a broad range of bacteria within a mixed population. The protocol involved a first PCR amplification using two different primer sets (one of which encompassing the V2, V4, V8 region, and the other the V3, V6-7, V9 one) for the amplification of hypervariable regions. The PCR program consisted of an initial denaturation of 95°C for 10 minutes, followed by 25 cycles of 95°C for 30 seconds, 58°C for 30 seconds, 72°C for 20 seconds, and a hold stage of 72°C for 7 minutes. Amplicons were quantified and pooled together to obtain a final concentration of 30 ng x µL$^{-1}$. The subsequent protocol involved the use of an Ion Xpress

**Table 1. Physico-chemical properties of the sampling sites. The data refer to the sampling dates presented in** Table 2**, except the water temperature (Mean Tw_max), whose value refers to the mean of the 3 weeks before the sampling, collected by dataloggers. Q: water discharge (absent in Agola being it a static pond and not a flowing kryal stream).**

| Sampling site | Agola (pond) | Mandrone (kryal) | Amola (kryal) |
|---|---|---|---|
| Altitude site (m asl) | 2596 | 2569 | 2540 |
| Distance from the source (km) | 0.02 | 0.05 | 0.14 |
| Suspended sediments (mg/l) | 2.60 | 76.80 | 131.00 |
| Mean Tw_max (°C) | 4.5±1.5 | 1.2±0.5 | 1.6±0.4 |
| Chlorophyll a (µg/cm2) | 0.09 | 0.48 | 0.33 |
| pH (20°C) | 8.20 | 6.9 | 6.6 |
| Conductivity (µS/cm) | 72.0 | 5.5 | 13.4 |
| SiO2 (mg/l) | 0.3 | 1.1 | 1.2 |
| Q (m³/s) | – | 3.8 | 1.1 |

Plus 9 Fragment Library Kit and an Ion Xpress Barcode Kit (Thermo Fisher Scientific, Waltham, MA, USA) for barcode ligation. A further amplification step was performed with the following program: 95°C for 5 minutes, 7 cycles of 95°C for 15 seconds, 58°C for 15 seconds, and 70°C for 1 minute, storing the amplified pools at 4°C. Libraries were quantified using a Qubit 3.0 Fluorometer with a Qubit™ DNA HS Assay Kit, in order to pool libraries at a final concentration of 100 pM. Samples were processed using an Ion 520™ and an Ion 530 ™ Kit – OT2 400 bp (Thermo Fisher Scientific Waltham, MA, USA) following the manufacturer's instructions. Samples were loaded on an Ion 520 chip and the sequencing run was performed in an Ion™ GeneStudio S5 System, both from the same manufacturer. Extractions and runs from negative controls, were performed as routine practice in compliance with recommended standards [35].

## Bioinformatics and statistics

Raw reads were processed to trim 20 base pairs on both ends to remove primers using cutadapt v3.4 [36] and analyzed using QIIME2 v2021.4 [37]. High-quality reads were denoised and dereplicated into amplicon sequence variants (ASVs) using the qiime dada2 plugin. To check if sequencing depth had been sufficient, an alpha-rarefaction plot was generated using the "qiime alpha-diversity" plugin. The SILVA SSU v138.1 database was used as a reference for the taxonomic assignment of ASVs [38] using the "classify-consensus-blast" plugin. The differential representation of ASV's between groups of samples was computed as log2 fold-change of the means. Chao1, Shannon-Wiener H value, Simpson's 1-D, and Community evenness (e^H/S) were computed from the output data matrix using Past 4.11 software. Alpha-diversity and evenness significance of differences were estimated by the MicrobiomeAnalyst online utility (https://www.microbiomeanalyst.ca/) using the Simpson, Shannon, or Chao1 Diversity ecological indicators. Before the analysis, the metagenomic data were filtered excluding reads with an abundance lower than 2 to minimize the effects of sequencing errors. The relative abundance % of the major taxa at genus level, and their differential representation analysis in the communities were analyzed using the SHAMAN online utility (https://shaman.pasteur.fr) using a three-variable model with the taxonomical depth set to the genus level.

Sequences have been deposited in NCBI's Sequence Read Archive under accession PRJNA1222183.

## Results and discussion

### Bacterial diversity of the glacier habitat is as high as that of temperate environments

Metabarcoding yielded 9,331,554 reads for the 48 samples after denoising, chimera removal, and quality filtering steps were applied. The mean number of reads per sample was 132,557.5, with a length range of 200–400 bases. Fig 2 shows the distribution of taxa (i.e., the number of different amplicon sequence variants, ASVs) and their cumulative read

**Table 2. List of the chironomid specimens and the environmental samples.** Taxa names for the insects were assigned by anatomical/morphological assessment supported by BLAST alignment of the sequenced cytochrome oxidase gene upon comparison with GenBank database (identification accepted upon observing scores of DNA sequence identity with named records higher than 99%). Top part: insects, bottom part: environment samples. SLPF is a sample from Prato Fiorito (side glacier of the Agola site).

| Site | date | Insect taxon | Type | Code |
|---|---|---|---|---|
| Mandrone | 03/09/2020 | *Diamesa steinboecki* | Larva | MADst1 |
| Mandrone | 03/09/2020 | *Diamesa steinboecki* | Larva | MaDst2 |
| Mandrone | 03/09/2020 | *Diamesa tonsa* | Larva | MADto3 |
| Mandrone | 03/09/2020 | *Diamesa tonsa* | Larva | MADto1 |
| Mandrone | 03/09/2020 | *Diamesa tonsa* | Larva | MaDto2 |
| Mandrone | 03/09/2020 | *Diamesa bertrami* | Larva | MADbe1 |
| Mandrone | 03/09/2020 | *Diamesa bertrami* | Larva | MADbe2 |
| Mandrone | 03/09/2020 | *Diamesa latitarsis* | Larva | MADla1 |
| Mandrone | 03/09/2020 | *Diamesa latitarsis* | Larva | MADla2 |
| Mandrone | 03/09/2020 | *Diamesa latitarsis* | Larva | MADla3 |
| Mandrone | 03/09/2020 | *Diamesa latitarsis* | Larva | MADla4 |
| Mandrone | 03/09/2020 | *Diamesa zernyi* | Larva | MADze1 |
| Mandrone | 03/09/2020 | *Diamesa zernyi* | Larva | MADze2 |
| Mandrone | 03/09/2020 | *Diamesa zernyi* | Larva | MADze3 |
| Mandrone | 03/09/2020 | *Diamesa zernyi* | Adult | MADzeA |
| Mandrone | 03/09/2020 | *Orthocladius* sp. | Adult | MAOrtA |
| Amola | 23/08/2017 | *Diamesa steinboecki* | Larva | AMDst1 |
| Amola | 23/08/2017 | *Diamesa steinboecki* | Larva | AMDst2 |
| Amola | 23/08/2017 | *Diamesa bohemani* | Larva | AMDbo1 |
| Amola | 23/08/2017 | *Diamesa bohemani* | Larva | AMDbo2 |
| Amola | 23/08/2017 | *Diamesa zernyi* | Larva | AMDze3 |
| Amola | 23/08/2017 | *Diamesa zernyi* | Larva | AMDze4 |
| Amola | 23/08/2017 | *Diamesa steinboecki* | Larva | AMDst |
| Amola | 23/08/2017 | *Diamesa bertrami* | Larva | AMDbe |
| Agola | 16/07/2018 | *Diamesa zernyi* | Larva | AGDze1 |
| Agola | 16/07/2018 | *Diamesa zernyi* | Larva | AGDze2 |
| Agola | 16/07/2018 | *Parametriocnemus stylatus* | Larva | AGPst1 |
| Agola | 16/07/2018 | *Parametriocnemus stylatus* | Larva | AGPst2 |
| Agola | 16/07/2018 | *Parametriocnemus stylatus* | Larva | AGPst3 |
| Agola | 16/07/2018 | *Metriocnemus eurynotus* gr. | Larva | AGMeu1 |
| Agola | 16/07/2018 | *Metriocnemus eurynotus* gr. | Larva | AGMeu0 |
| Agola | 16/07/2018 | *Metriocnemus eurynotus* gr. | Larva | AGMeu2 |
| Agola | 16/07/2018 | *Pseudokiefferiella parva* | Larva | AGPpa1 |
| Agola | 16/07/2018 | *Pseudokiefferiella parva* | Larva | AGPpa2 |
| Agola | 16/07/2018 | *Pseudokiefferiella parva* | Larva | AGPpa3 |
| Agola | 16/07/2018 | *Diamesa latitarsis* | Larva | AGDla0 |
| | | **Habitat/Environment samples** | | |
| Agola | 20/07/2021 | Sediment mud in proglacial pond south side | Pond | SMAG1 |
| Agola | 20/07/2021 | Slime on submerged stones proglacial pond south side 1 | Pond | SLAG1 |
| Agola | 20/07/2021 | Sediment mud in proglacial pond North side | Pond | SMAG2 |
| Agola | 20/07/2021 | Slime on submerged stones proglacial pond south side 2 | Pond | SLAG2 |
| Agola | 20/07/2021 | Soil under glacier ice | Glacier | ICAG |
| Agola | 20/07/2021 | Water percolating from melting ice over glacier | Glacier | BdWAGa |

*(Continued)*

**Table 2.** (Continued)

| Site | date | Insect taxon | Type | Code |
|------|------|--------------|------|------|
| Agola | 20/07/2021 | Organic detritus in snowy portion over glacier ice | Glacier | ORAG |
| Agola | 20/07/2021 | Organic detritus in eroded glacier bottom rivulet | Glacier | ORAGh |
| Agola | 20/07/2021 | Ice core drilled with pickaxe from glacier side | Glacier | BdWAGb |
| Agola/P.F. | 21/07/2021 | Pond stream in nearby Prato Fiorito ice formation | Pond | SLPF |
| Agola | 20/07/2021 | Melting ice brooklet flowing fromresidual glacier core | Glacier | BdWAGx |
| Agola | 20/07/2021 | Water from proglacial pond | Pond | WAAG |

abundance in each sample. The first notable finding is that the samples from the environment (both the proglacial pond and the glacier itself) cluster together at the top of the dendrogram. These samples are labelled in light blue and feature values of richness averaging 4266, ranging from a minimum of 2,215 to a maximum of 7,641. This is consistent with levels recorded in various temperate soil environments, including agricultural soils under organic or conventional management [39] and soil recovery following degradation and reconstitution trials [40]. In those two studies, we used the same sequencing method (ION S5 multi-amplicon) and bioinformatics pipeline.

The possible interpretation of a vast microbial diversity in a harsh high-mountain environment is that it could be possibly widely contributed by cell immigration through passive atmospheric discharge rather than by in situ deterministic processes. Bacterial deposition rates across the earth have been shown to range from $0.3 \times 10^7$ to $8 \times 10^7$ cells/m²/day [41]. The estimated global emission rates of bacteria that fall back on our planet amount between 0.7–28.1 Tg/year [42]. Airborne microbiota have been found to be less affected by sampling location than by all other variables [43,44].

### Insect-associated bacterial communities have far lower diversity than those of their habitat, and minimal species overlap with them

A central piece of evidence that addresses one of the key questions of the present project comes from comparing the species diversity of chironomid-associated bacterial communities with that of their surrounding environment. As Fig 2 shows:

a) The number of taxa extracted from the insect samples is, on average, almost sevenfold lower than that of the glacier or pond samples;

b) All insect communities end up clustered separately from all environmental ones in the neighbour-joining dendrogram, which is constructed based on Bray–Curtis beta diversity distances. The dendrogram partitions into two main clusters, one of which (occupying the mid-position in the Fig 2 tree) has visibly higher diversity than the other. It can also be seen that this subgroup has a correspondingly higher abundance of individuals in its samples, yielding higher sequence reads. This in turn enables the detection of rarer taxa and allows more similarities to be traced, which eventually shape the tree topology, as the image shows. This correlation between read abundance and detectable alpha diversity is common in metabarcoding studies.

Two adult specimens belonging to two different chironomid species were found in the Mandrone glacier. The adults, which are rarely encountered, were included primarily because they were available and provided a form of control. They do not live in contact with the same aquatic environment in which the larvae are found; they hardly fly unless there is no wind and tend not to feed or feed on very different sources, such as plant pollen. They are very short-lived and usually die soon after mating. The two specimens were included in the analysis to see if their microbiome would display any particular differences, although this could not be statistically supported. Some authors have reported an expected decline in microbiome diversity throughout the developmental stages from egg to larva and to adult in *Chironomus transvalensis* [45].

 

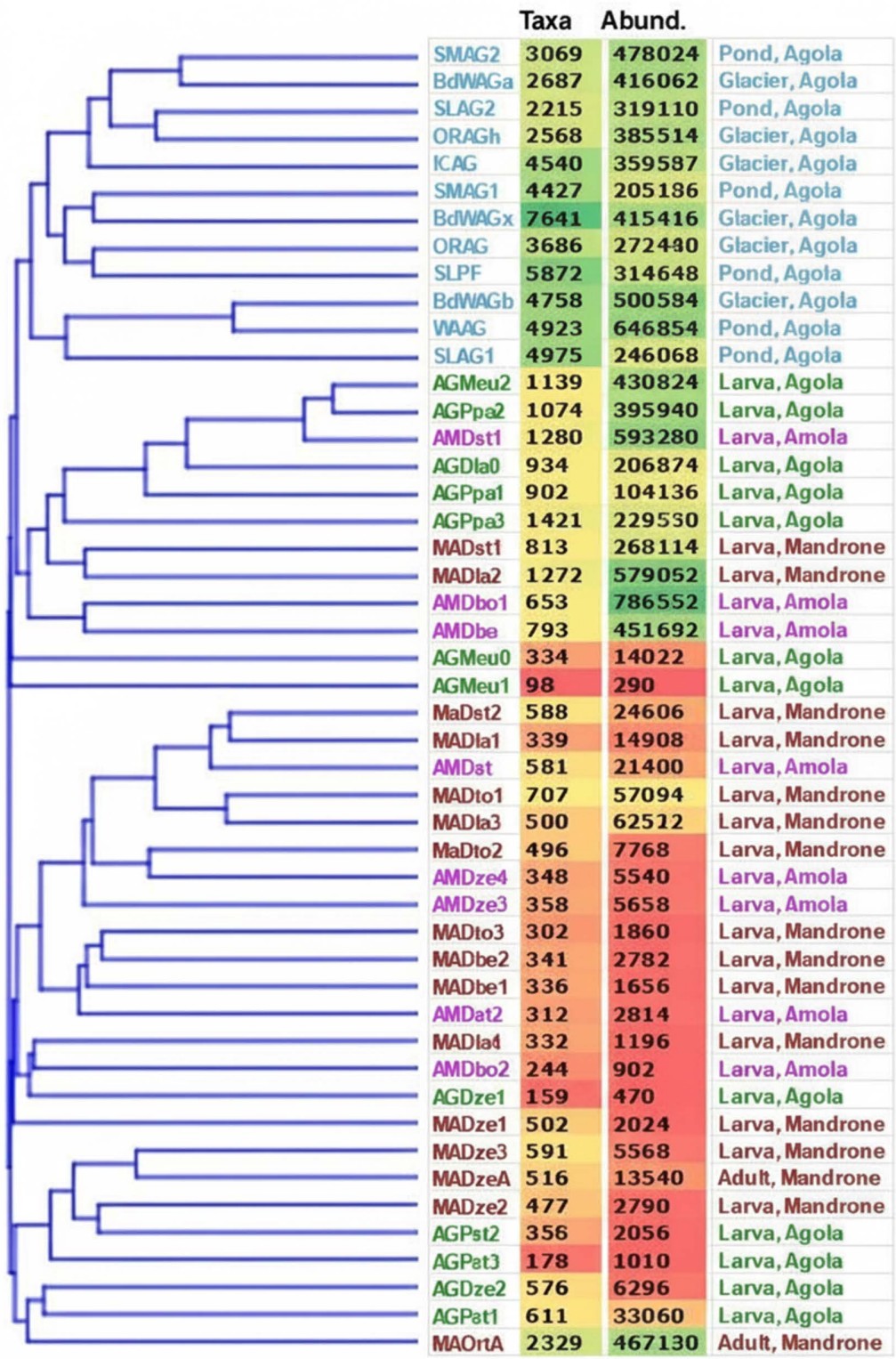

**Fig 2. Cluster Analysis results.** Neighbor-Joining dendrogram based on the Bray-Curtis intercommunity distances. The number of different taxa, expressed at the highest resolution level (Amplicon Sequence Variants, ASV), is shown along with the number of reads (Abund.) occurring in each sample. Codes and descriptions in light blue are from the environmental samples (glacier or proglacial pond). For the other colored text fonts, the following

coding applies; green: Agola glacier, purple: Amola glacier, brown: Mandrone glacier. The cells reporting the numbers of taxa and the reads abundance are colored by the Microsoft Excel conditional formatting gradient from red (lowest numerical values) to green (highest numerical values).

In our results, the positions of the two adult samples in the microbial community dendrogram showed no signs of self-similarity, and their alpha diversities were also very different from each other.

Regarding the lower diversity of insect-associated bacteria compared to their external environment, this is consistent with existing evidence: animal gut microbiomes, including the human microbiome, comprise hundreds of species, whereas general open environments account for thousands [46,47].

The observation that chironomids have fewer sequence variants than their surrounding environment is indeed not contrary to ecological knowledge. There is compelling evidence to support the idea that many detritivorous insects harbour a limited set of gut symbionts and that the low bacterial diversity in their guts, compared to the surrounding environment, reflects the selective maintenance or filtering of specific microbial communities. Insect guts often contain fewer microbial species than mammalian guts, and many insects rely on specialized bacterial communities for essential functions such as digestion and defense [19]. A large-scale study analyzing gut microbiota across 218 insect species found that gut bacterial diversity is significantly lower than environmental microbial diversity and that omnivorous and detritivorous insects tend to host fewer bacterial taxa. The composition of gut microbiota resulted shaped by host diet, developmental stage and phylogeny, thereby reinforcing the role of host-driven selection [48].

## The hierarchical ranking of the variables in shaping microbiota indicates as insignificant both the insect species and the geographical site

Another key finding of the multivariate clustering ordination is that, as the tree clearly shows, the external habitat was the dominant driver of bacterial community structure. In contrast, the chironomid species and the glacier from which the samples were collected have little appreciable effect. The two clusters into which the chironomid sample communities fall (Fig 2) show no evidence of insect species coherence in the formed sub-branches. Likewise, the origin from the Agola, Amola or Mandrone glaciers; reveal no site-driven clades.

Thus, regarding the possible weight of the variables, i.e., insect species and the geographical location of the glaciers, the host or site effects, if any, were minimal, as samples from the Mandrone, Agola, and Amola glaciers were essentially mixed together across the branches of the multivariate cluster analysis tree. This profound difference in the strength of the variables' influence and the magnitude of the gaps between them has actually been observed repeatedly in metabarcoding studies.

The evidence mentioned above of little overlap between habitats and larval communities, and the higher species richness of the former, is better summarized in the two panels of Fig 3. The relative abundance of the dominant genera in the glacier and pond samples, and in all insect samples, shows that the glacier and its proglacial pond are very similar, but rather distinct from the bacterial community structure of the insects in terms of the overall proportions of the main microbial taxa (Fig 3a). Furthermore, the alpha diversity ecological indices confirm that the insect communities are less diverse, as indicated by the lower Chao1 and Shannon values compared to the high scores of the two types of habitat (Fig 3b).

For the alpha diversity comparison (Fig 3b), an overall statistical difference was found following a Welch ANOVA test, giving an F-value of 42.75 and a highly significant p-value of $3.945e^{-11}$. The Wilcoxon Rank Sum post-hoc test revealed the following differences: insect vs. pond: 0.011061; insect vs. glacier: 0.014671; pond vs. glacier: 0.8598. The p-values were calculated using the Benjamini–Hochberg false discovery rate correction.

To provide further statistical evidence for the separation between insect-associated microbiota and the surrounding environment, we performed a multivariate beta-diversity analysis using non-metric multidimensional scaling (NMDS), the plot of which is presented in Fig 4. Three statistical assessment approaches were also performed (PERMANOVA, ANOSIM and PERMDISP), all of which scored significance for the difference between samples.

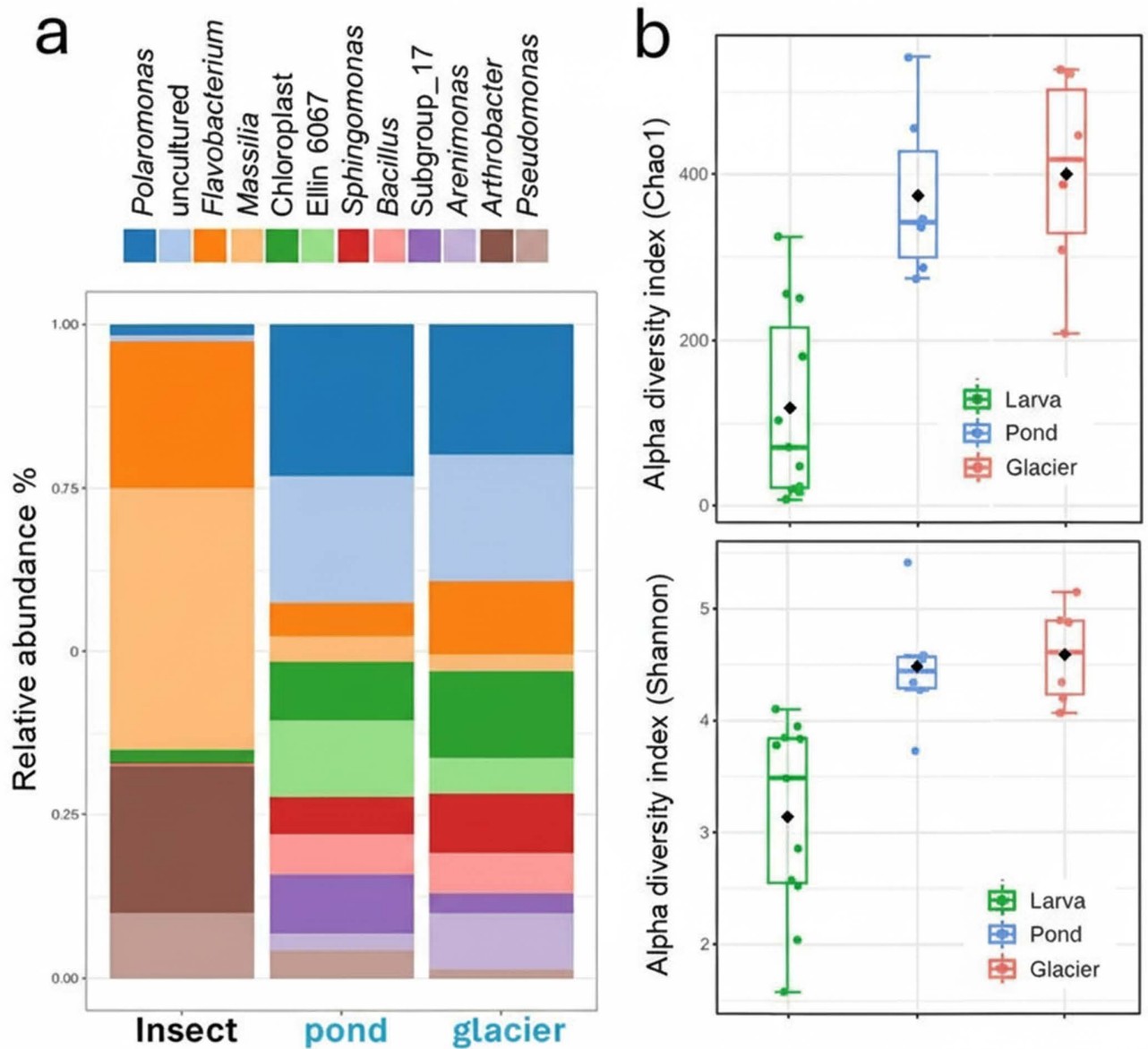

**Fig 3. Main genera and alpha diversities.** a) Histogram bar representation of the comparative percentage abundances of the 12 most prevalent genera when all insect sample data are pooled and compared with the pond and glacier datasets. b) Alpha diversity comparisons for the same subsets of data using the two ecological indexes: Chao 1 (top panel) and Shannon diversity (bottom panel).

## Hierarchical ranks partitioning proportions rule out a passively accumulative feeding behaviour

Another key question this investigation aimed to address was the extent to which bacteria in the surrounding environment can serve as a direct food source for chironomid larvae living in glacier habitats and consequently foraging on existing organic resources. The ingestion of cells by animals would bring the corresponding DNA into the consumer's gut, where it could in principle be detected by metabarcoding analyses. This issue is also relevant to the possibility of determining an animal's dietary composition by analyzing the DNA sequences extracted from its digestive tract. The hypothesis tested was that the food web in glacier waters, including the ice interface and proglacial pond, could substantially include live

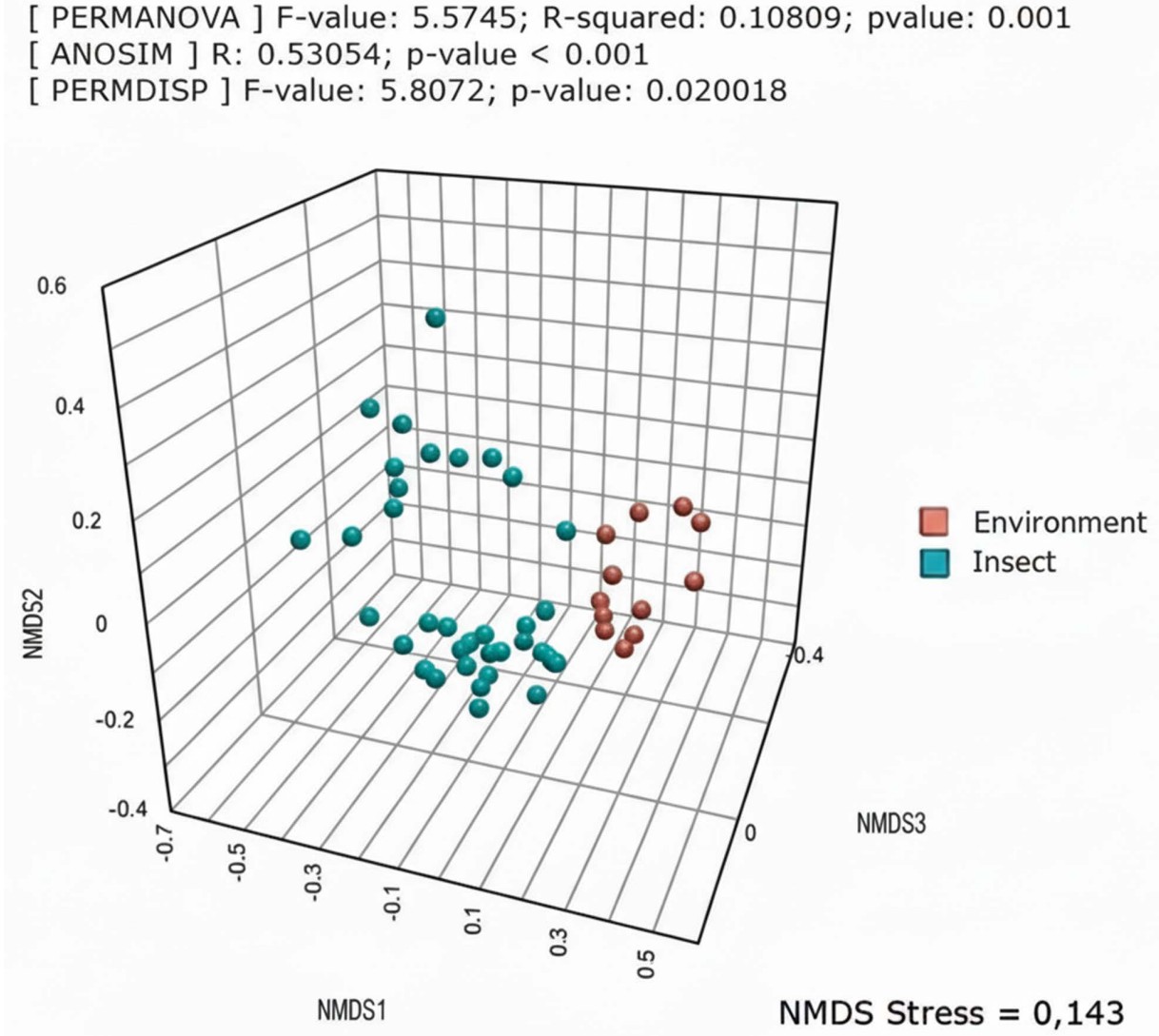

[ PERMANOVA ] F-value: 5.5745; R-squared: 0.10809; pvalue: 0.001
[ ANOSIM ] R: 0.53054; p-value < 0.001
[ PERMDISP ] F-value: 5.8072; p-value: 0.020018

NMDS Stress = 0,143

**Fig 4. Beta diversity and statistical differences between pools.** 3-D NMDS plot showing the separation of bacterial communities in insect samples from those in the surrounding glacial environment. The results of the permutational multivariate analysis of variance (PERMANOVA), permutational analysis of multivariate dispersions (PERMDISP) and analysis of mean rank similarities (ANOSIM), all based on Bray–Curtis intercommunity distances, are reported over the graph.

bacteria as part of the chironomid larvae's nutrient and energy supply. In this case, extracting total DNA from the larval body and sequencing the amplicons produced using 16S primers for prokaryotes allows to compare the degree of identity with the corresponding sequences obtained from the external environment and count the co-presence within the two groups. The data from such a comparative assessment are shown in Table 3.

The table shows the collective results of bacterial amplicon sequencing. The first five columns from the left (from phylum to species) stem from the resulting bioinformatic annotations that match the available records in public databases by alignment. The last column on the right (ASVs) shows the number of different sequence variants yielded by this metabarcoding. It should be noted that the species-level records in public databases are still incomplete for this type of metagenomics experiment, as caution is rightly exercised when attributing identities to this finer rank or coining new

**Table 3. Differential taxonomy structures. Number of different taxa observed across the different rank levels for the entire project or in subsets of samples, distinguishing between those occurring only in habitat samples, only insect samples, or shared by both.**

|  | n. phyla | n. orders | n. families | n. genera | n. species | n. ASVs |
|---|---|---|---|---|---|---|
| Whole analysis | 47 | 358 | 594 | 1264 | 345 | 52336 |
| Only in habitat | 6 (12.77%) | 103 (28.77%) | 174 (29.29%) | 431 (34.10%) | 145 (42.03%) | 36742 (70.20%) |
| Only in insect | 2 (4.26%) | 14 (3.91%) | 39 (6.57%) | 186 (14.72%) | 116 (36.62%) | 13016 (24.87%) |
| In both | 39 (82.98%) | 241 (67.32%) | 381 (64.14%) | 647 (51.19%) | 84 (24.35%) | 2578 (4.93%) |

species names. Therefore, while the number of possible species in nature is, by definition, higher than that of genera (or at least equal), metabarcoding annotations normally show the reverse, as the algorithm prudently leaves the species rank undetermined.

Inspecting the percentage of shared taxa from left to right along the taxonomic resolution gradient shows that the proportion of shared taxa between the outside environment and insect-associated bacterial DNA decreases progressively, culminating in a minimal conserved core of just 4.93% at the finest resolution level, consisting of single amplicon sequence variants (ASVs). The community composition is therefore revealed to consist of a majority (>70%) of taxa that remain exclusively in the environmental pool, as well as a significant proportion (almost 25%) that are strictly insect-specific, as they are not encountered in the glacier pond. These taxa mostly entail the gut microbiome of chironomid juveniles. The answer to the original question that excludes the hypothesis of a gut content mirroring the available resource community is better visible in the pie charts progression reported in Fig. S1 in S1 File.

The above datum for shared variants, i.e., less than 5% similarity is objectively a low proportion. If microbial flow from the environment into the consumer were a direct reflection of the external assemblage, the dominant members of both communities would be the same, ruling out coincidental effects.

The progressive reduction in the proportion of taxa observed in both the external environment and insects, as taxonomic resolution increases, suggests in fact an underlying evidence: diversity outside is overwhelmingly higher than diversity inside. If the process were merely an accumulative feeding, one would not see this outcome, but rather a nearly identical picture between the two pools. Even feeding behaviour involving selective fractionation of ingested cells (unlikely given the size of the 'prey' in comparison to the larval foragers) would lead to reduced diversity but consistent identity with the source of the cell pools. Furthermore, the high proportion of taxa found only in insects shows that the chironomid microbiome is essentially animal-specific, suggesting a functional role rather than merely acting as a reservoir for communication between the outside and the inside.

Regarding the suitability of inferring diet content from animal gut microbiomes, it should also be considered that rapid digestion or fast DNA degradation may influence the observed results. However, in studies with larger animals plant DNA is usually retrievable from the faeces of herbivores [49].

To provide additional, independent assessments using different ecological indices, and to uncouple the differences between the bacterial assemblages in the pond and glacier with respect to the insects, we performed two further calculations. The Jaccard similarity index, computed from the mean of each of the three categories of the whole ASV data matrix, yielded the following values: insect vs. pond: 0.0564; insect vs. glacier: 0.0548; glacier vs. pond: 0.1336. The Bray–Curtis distances were as follows: insect vs. pond: 0.9367; insect vs. glacier: 0.9412; glacier vs. pond: 0.5504).

The data show a high degree of similarity and low distance between the bacterial communities of the two environmental samples (glacier vs. pond), and an opposite situation for each of those communities when compared with the insect category.

To inspect the coherence of the output from a single glacier with that from the three altogether, we compared the percentages of phyla and ASVs that occurred in the habitats only, those shared between the habitats and insects, and those occurring in insects only, for the different datasets of single glacier versus all. The resulting trends were very consistent, as shown in Supporting Information Fig. S2 in S1 File.

Additionally, a Venn diagram showing the number of shared vs. specific ASVs in glacier, pond, and insect samples is provided in the Supplementary Information (Fig. S3) in S1 File.

## Dominant taxonomical names recurrence sets strongly apart the insect communities from the habitat ones

Metagenomics sequencing outputs also offer the possibility of extracting another dimension from the data relating to the relationships between bioinformatically assigned taxonomic names and primary nucleotide sequence variants. We previously outlined this and other possibilities for metabarcoding datasets from a series of environments [50,51]. In this respect, in addition to the relative abundance of sequence reads leading to a given taxonomic name (on which the results shown in the above figures and tables are based), one can compute the frequency with which each taxonomic lineage is pointed out by each sequence variant (by creating a matrix in pivot tables) and compare this type of output. This is particularly useful for evaluating and comparing community structures in subsets of the experimental dataset by listing the respective scores in order of decreasing abundance. Table 4 shows excerpts from the top of the list for the two subsets of data relating to lineages observed only in insect-associated DNA samples and those found exclusively in the glacier/pond environment.

Two main pieces of evidence can be seen in the compared lists. Firstly, in the insect-specific pool, the most recurring annotations account for up to 7.86% of the total, and all the names in the top-scoring taxa list exceed 1%. Conversely, none of the members of the environmental list exceed 0.8%, indicating a lower level of oligarchy/dominance in this external community. 2) The most recurring names in each list differ, with the insect core group featuring different variants of the *Providencia*, *Serratia*, *Massilia* and *Flavobacterium* genera. In contrast, the adjacent environment, in which those larvae thrive, is characterized by a more even array, including an undetermined member of the Vicinamibacterales order, the Nitrosomonadaceae family, and the *Arenimonas*, *Methylotenera*, *Rhizobacter* and *Polaromonas* genera. The presence of *Providencia* and *Serratia* associated to chironomids is in agreement with prior evidence [22,23] Essentially, the structure and taxa composition highlight the profound differences between these two sets of bacteria.

Concerning the taxa encountered and differentially represented in the chironomids or glacier environments, *Serratia* is often associated with insects, including the dipteran *Bactrocera dorsal*is, and can exhibit symbiotic properties [52,53]. *Providencia*, including *P. stuartii*, which is among the annotated results for this group, is a reported insect pathogen [54]. *Massilia* is a taxon with a high level of enzymatic versatility and includes psychrophilic species [55]. *Undibacterium* is an aquatic degrader of polyhydroxyalkanoates [56]. *Flavobacterium psychrophylum* is the agent of cold-water disease in salmonid fish [57]. *Deefgea* is a chitinolytic dweller of fish guts [58]. The *Alkanindiges* genus is an obligate hydrocarbonoclastic aliphatic hydrocarbon degrader [59]. This taxon is also featured in glacier data, as is *Polaromonas*, a typical inhabitant of glacier environments [60]. *Arenimonas* and *Methylotenera* have both been found in active glacier consortia [61]. *Rhizobacter*, on the other hand, was isolated from freshwater sediments [62].

## Taxa that occur uniquely in the habitat or in association with insects show peculiarities at all rank levels

To thoroughly extract the characteristic bacteria that distinguish the two niches, Table 5 compiles the lists of those taxa, which, at different ranks, are encountered not just as differentially represented in their frequencies but as totally unique.

In such a comparison, it is important to consider that the concept of uniqueness is a result defined by the detection level. A low-abundance taxon could be enriched in one compartment (e.g., insect or habitat) but be too rare to be detected in samples where it would be underrepresented. Thus presences and absences would not necessarily be due to specificity or exclusion but possibly also to disproportionate numbers in one of the two groups that could make a taxon technically undetectable.

Bearing this caveat in mind, the data nevertheless reveal several major differences. As can be seen in Table 5, even at the highest taxonomic level (phylum), two phyla are exclusively encountered in insects and six are only found in their habitats. Similarly, many orders separate the two sets, followed by proportionally longer lists of families and genera. The table shows those that reach the highest quantitative representation in the sequencing outputs. These data further emphasize that specificity, rather than symmetry, defines the two community types.

**Table 4. Dominant bacterial nomenclature in habitat or insects. Top-scoring excerpts in order of decreasing percent abundance (first column), showing the most frequently occurring names from the two lists of lineage outputs in the pivot tables assigned to the ASV data matrices. Note that while a given ASV with an assigned lineage may be unique to either insects or the habitat, this does not preclude the possibility that a different ASV bearing the same taxonomic lineage could be common to both (for completely unique taxonomic names, refer to Table 5. in the subsequent paragraph).**

**Taxonomy annotations of the top most recurring taxa names for ASVs found <u>only in insects</u>**

| % | Class | Order | Family | Genus |
|---|---|---|---|---|
| 7,866 | Gammaproteobacteria | Enterobacterales | Morganellaceae | *Providencia* |
| 6,799 | Gammaproteobacteria | Enterobacterales | Yersiniaceae | *Serratia* |
| 5,899 | Gammaproteobacteria | Enterobacterales | Yersiniaceae | *Serratia* |
| 4,817 | Gammaproteobacteria | Enterobacterales | | |
| 3,222 | Gammaproteobacteria | Enterobacterales | Yersiniaceae | *Serratia* |
| 3,062 | Gammaproteobacteria | Enterobacterales | Morganellaceae | *Providencia* |
| 2,352 | Gammaproteobacteria | Enterobacterales | Yersiniaceae | *Serratia* |
| 1,915 | Actinobacteria | Micrococcales | Microbacteriaceae | |
| 1,529 | Gammaproteobacteria | Burkholderiales | Oxalobacteraceae | *Massilia* |
| 1,485 | Bacteroidia | Flavobacteriales | Flavobacteriaceae | *Flavobacterium* |
| 1,428 | Actinobacteria | Micrococcales | Microbacteriaceae | |
| 1,267 | Gammaproteobacteria | Burkholderiales | Oxalobacteraceae | *Massilia* |
| 1,240 | Gammaproteobacteria | Burkholderiales | Oxalobacteraceae | *Undibacterium* |
| 1,128 | Gammaproteobacteria | Burkholderiales | Oxalobacteraceae | *Massilia* |
| 1,109 | Actinobacteria | Micrococcales | Micrococcaceae | *Arthrobacter* |
| 1,066 | Gammaproteobacteria | Enterobacterales | Yersiniaceae | *Serratia* |
| 1,061 | Gammaproteobacteria | Burkholderiales | Oxalobacteraceae | *Massilia* |
| 1,004 | Bacteroidia | Flavobacteriales | Flavobacteriaceae | *Flavobacterium* |

**Taxonomy annotations of the top most recurring taxa names for ASVs found <u>only in habitats</u>**

| 0,820 | Vicinamibacteria | Vicinamibacterales | uncultured | uncultured |
|---|---|---|---|---|
| 0,726 | Gammaproteobacteria | Burkholderiales | Nitrosomonadaceae | Ellin6067 |
| 0,641 | Gammaproteobacteria | Xanthomonadales | Xanthomonadaceae | *Arenimonas* |
| 0,632 | Thermoanaerobaculia | Thermoanaerobaculales | Thermoanaerobaculaceae | Subgroup10 |
| 0,554 | Bacilli | Bacillales | Planococcaceae | Soli *Bacillus* |
| 0,534 | Gammaproteobacteria | Burkholderiales | Nitrosomonadaceae | Ellin6067 |
| 0,512 | Acidimicrobiia | Microtrichales | Ilumatobacteraceae | CL500-29marine |
| 0,503 | Gammaproteobacteria | Burkholderiales | Methylophilaceae | *Methylotenera* |
| 0,491 | Gammaproteobacteria | Burkholderiales | Comamonadaceae | *Rhizobacter* |
| 0,470 | Thermoanaerobaculia | Thermoanaerobaculales | Thermoanaerobaculaceae | |
| 0,418 | Acidimicrobiia | Microtrichales | Ilumatobacteraceae | CL500-29marine |
| 0,400 | Gammaproteobacteria | Burkholderiales | Comamonadaceae | *Rhizobacter* |
| 0,393 | Gammaproteobacteria | Burkholderiales | Comamonadaceae | *Polaromonas* |
| 0,380 | Gammaproteobacteria | Burkholderiales | Nitrosomonadaceae | MND1 |
| 0,377 | Gammaproteobacteria | Burkholderiales | Comamonadaceae | *Rhizobacter* |
| 0,374 | Alphaproteobacteria | Rhodobacterales | Rhodobacteraceae | *Rhodobacter* |
| 0,322 | Cyanobacteriia | Leptolyngbyales | Leptolyngbyaceae | *Chamaesiphon* |

Regarding the lineages that were uniquely encountered and are shown in Table 5, the following comments can be made on their identities. The archaeal phylum Crenarchaeota has been found in cold environments, including retreating glacier soils [63]. They are known for ammonia oxidation and are often detected in environmental DNA surveys of cold

**Table 5. Uniquely occurring taxa in insects and environment.** Lists of the unique cases, i.e., detected specifically only in insect or in the habitat samples, for the different taxonomic ranks.

| | Unique for Insects | Unique for environment |
|---|---|---|
| **Phyla** | Crenarchaeota, Margulisbacteria | Sumerlaeota, WS4, WS2, Hydrogenedentes, Caldisericota, Dadabacteria |
| **Orders** (top 20 most abundant for habitats) | Aneurinibacillales, Oligosphaerales, Haloplasmatales, Thermicanales, Competibacterales, Nitrososphaerales, Thermovenabulales | Vibrionales, Syntrophomonadales, Paracaedibacterales, Endomicrobiales, Phormidesmiales, Dadabacteriales, Caldisericales, Sumerlaeales,_Anaerolineales, Parvibaculales, Euzebyales, Moorellales, Defluviicoccales, Latescibacterales, Syntrophorhabdales, Ardenticatenales, Entomoplasmatales,_Methylacidiphilales, Thermodesulfovibrionales, Thiomicrospirales |
| **Families** (top 20 most abundant) | Fusibacteraceae, Aquaspirillaceae, Jonesiaceae, Aneurinibacillaceae, Arcobacteraceae, Chromobacteriaceae, Muribaculaceae, Symbiobacteraceae, Sporolactobacillaceae, Oligosphaeraceae, Thermicanaceae, Xenococcaceae, Thioalkalispiraceae, Haloplasmataceae, Ignavibacteriaceae, Beutenbergiaceae, Competibacteraceae, Ferrovaceae, Nitrososphaeraceae, Thermomicrobiaceae | Vibrionaceae, Smithellaceae, Syntrophomonadaceae, Paracaedibacteraceae, Endomicrobiaceae, Lentimicrobiaceae, Nodosilineaceae, Caldisericaceae, Sumerlaeaceae, Idiomarinaceae, Trueperaceae, Hydrogenedensaceae, Anaerolineaceae, Elsteraceae, Parvibaculaceae, Helicobacteraceae, Tannerellacea, Euzebyaceae, Oscillatoriaceae, Moorellaceae |
| **Genera** (top 20 most abundant) | *Providencia, Pectobacterium, Geminocystis*_PCC-6308, *Thermobacillus, Xenorhabdus, Yokenella, Leptotrichia, Veillonella, Fusibacter, Methyloversatilis, Symphothece*_PCC-7002, *Laribacter, Thermobispora,* [*Agitococcus*]_*lubricus*_group, *Bordetella, Geobacillus, Phascolarctobacterium, Hydrogenophilus, Jonesia, Trichococcus, Terrabacter, Aneurinibacillus* | *Sulfuricella,* [*Ruminococcus*]_*torques*_group, *Vibrio, Smithella, Tahibacter, Perlucidibaca, Syntrophomonas, Aliterella, Endomicrobium, Erysipelatoclostridium, Candidatus*_Paracaedibacter, *Fusicatenibacter, Nodosilinea*_PCC-7104, *Caldisericum, Butyricicoccus, Clostridium*_sensu_stricto_18, *Kazania, Sumerlaea, Idiomarina, Pseudorhodoplanes* |

habitats. Margulisbacteria are siblings of Cyanobacteria with a hydrogen-based metabolism and can be found in harsh environments [64]. Sumerleota and. Hydrogenedentes are associated with the subsurface biosphere, are involved in sulfur and nitrogen cycling, and are adapted to anoxic conditions [65]. A similar case can be made for Caldisericota, which are more thermophilic and anaerobic, and are found in geothermal and subsurface environments, but have also been signalled in glacier environments [66]. At the genus level, *Providencia* has also been retrieved from insect guts as an opportunistic pathogen and in aquatic environments [67]. *Veillonella* is an anaerobic taxon that is found in gut microbiomes including insects [68]. *Leptotrichia* is a host-associated bacterium found also in the microbiomes of insects [69]. *Yokenella* is encountered also in aquatic insects and is gut-associated [70]. *Xenorhabdus* is reported to be symbiotic with nematodes, but pathogenic in insect guts [71]. *Phascolarctobacterium*, instead, is reported to be a succinate fermenter in insect-resident microbiota [72]. Many of the habitat-specific genera are ecologically coherent, playing roles in sulphur cycling, and syntrophy. They are often free-living in glacial environments and include *Sulfuricella* [73], *Smithella* [74]. Extremophiles as *Thermobacillus* [75] or *Geobacillus* [76] and *Hydrogenophilus* [77] are instead reported as cases of thermophiles that are nevertheless found within cold contexts as permafrost, occurring also in Antarctica and Himalaya. Other taxa that are only found in environmental samples are known to be cyanobacteria, which can contribute to primary production in water and epilithic habitats. These include *Geminocystis* [78], *Symphothece* [79], *Nodosilinea* [80] and *Aliterella* [81].

### The minority of taxa shared in both insect and habitat communities are mostly overrepresented in insects

Regarding the 4.93% of sequence variants (2,578 cases; Table 3) that were observed in both insects and the glacier or pond environments, we calculated the mean for each across the insect samples and the environment samples. We then calculated the extent of the $\log_2$ fold change in one set of samples versus the other. This enabled to assess how many of the variants would feature an increase or a decrease. The extent of change had a maximum value of 14–15-fold $\log_2$ changes. The comparison is shown in Table 6.

**Table 6. Differential abundance proportions of the shared variants.** Summary of the behaviour of the 2578 sequence variants observed in both insects and environmental samples. The table specifies how many of these variants showed an over-representation (i.e., a $\log_2$ fold-change in the mean) higher in the insect sample compared to the environmental sample, or vice versa.

| | Number of ASVs | Percent over the total of shared ASVs |
|---|---|---|
| log2 fold-change of ASV mean unchanged | 14 | 0.54% |
| log2 fold-change of ASV mean decreased in insects | 359 | 14.31% |
| log2 fold-change of ASV mean increased in insects | 2195 | 85.14% |

The vast majority of comparisons revealed that these shared taxa (85.14%) are overrepresented in insects. The mean $\log_2$ fold change in these taxa was 2.28. For the smaller proportion (14.31%) that decreased in insects, the mean $\log_2$ fold change was −4.36. This evidence further supports the idea of a direct effect related to the insect host, rather than passive intake, as outlined in the discussion comparing this table with the data shown in Table 3.

Regarding these results they suggest further considerations. As Table 3 shows, overall, ASVs are richer outside (i.e., in the habitat, at 70.20%) than inside chironomids (at 24.87%). However, when an ASV is shared (a rare event, occurring in 4.93% of cases), its abundance counts across the larvae samples are higher than those displayed by the habitat samples (Table 6). This suggests that the variants that 'belong' to the insect are enriched in them despite the surrounding situation If it were merely a matter of passive intake, since the habitat is richer in general diversity, we would not see its side 'relatively depleted' for the bacteria that are shared.

Considering that the 12 habitat samples have a mean number of taxa (ASV) of 4266.7, while the 36 insect samples have a richness that is almost seven times lower (mean = 633.05), if these phenomena were passive, the probability of an ASV being in the environment would be seven times higher than the probability of it being in a chironomid inhabiting the environment. Conversely, as shown in Table 6, examining the $\log_2$ fold changes of the shared sequence variants reveals a nearly symmetrically reversed situation: a sixfold higher abundance of shared ASVs (85.14% vs. 14.31%), and a mere 2.8-fold higher presence of distinct ASVs in the 'habitat only' set than in the 'larvae only' set (70.20% vs. 24.87%, Table 3).

### Instances of different sequence variants sharing the same taxonomic name reveal microevolutionary trends specific to either insects or habitats

Another intriguing observation from the datasets was that, in most cases, different individual sequence variants that ended up being annotated with the same nomenclatural lineage at all ranks tended to be found in either insect or habitat samples. This is essentially what we previously noted in Table 3 onwards, but with an important additional aspect. That is, despite the difference in the 'oligarchic' dominant taxa of the top scores (shown in Table 4), these ASV-based strong differences in insect vs. environmental bacterial community composition come from datasets that contain a very high number of different variants, which, based on our current microbiological knowledge, are classified under the same name.

This suggests that, despite having a common ancestry origin, what we call under the same taxonomic name has nevertheless diverged in its exact sequence in relation to whether it is insect-borne or free-living in the environment.

In other words, most ASVs that are taxonomically named in the same way (at the genus or even species level) are different sequence variants that also show traces of microevolution. To illustrate this phenomenon, which we found widespread in our dataset, we present two examples from the analysis spreadsheets, selected from taxa that have been attributed to the deepest taxonomic level (i.e., species), for which several ASVs have converged to the same classification. The chosen examples are *Flavobacterium psychrophilum* and *Pantoea agglomerans,* whose details are shown in Fig 5.

As the image shows, the same microbial species can be found in glacier biotic communities either as variants that occur only in insects, only in environmental samples, or as shared variants. All these variants have differences in their

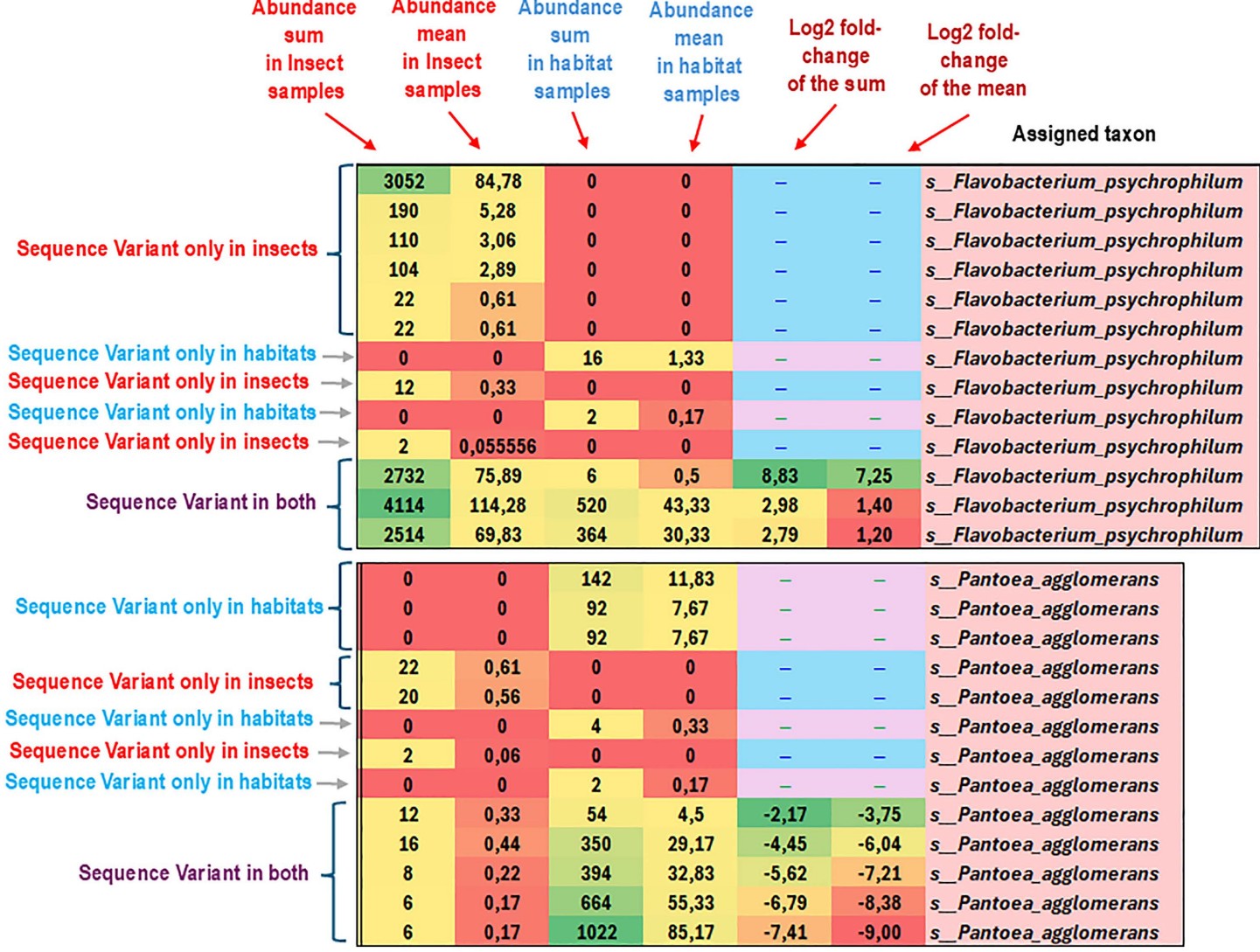

**Fig 5. Microevolutionary intraspecies clues.** This image is a detail of a spreadsheet matrix showing the results of single ASVs. It highlights examples of taxa that are classified under the same species name, and it shows microevolutionary clues of specialization towards insects (*F. psychrophilum*) or the environment (*P. agglomerans*). There are also some intermediate cases, but their proportions are consistent with the prevailing trends of the specialized ones. The green-to-red colour coding of the conditional formatting is used as described in the legend of Fig 2.

ribosomal RNA gene nucleotide sequence, which forms the basis for metabarcoding and constructing a tree of life. In each of these cases, the sequence still has enough base pair identities to lead to the same bacterial species name. However, it has undergone a number of variations/mutations in the analyzed mountain sites, giving rise to at least 13 variants for both *P. agglomerans* and *F. psychrophilum*.

## Sequence variants that are found in both the insect and the environment have a higher bioinformatic identification success rate

This aspect concerns the interesting subset of variants encountered in both insect and environmental samples, as opposed to those found in only one of the two categories. In metagenomics, due to the incompleteness of the databases,

| | Total ASVs | ASVs only in habitat | ASVs only in insects | ASVs only in both |
|---|---|---|---|---|
| n. of ASVs | 52336 | 36742 | 13016 | 2578 |
| n. of taxonomy names | 2780 | 2321 | 1662 | 555 |
| n. ASVs per name | 18,8 | 15,8 | 7,8 | 4,6 |
| taxonomy names % of ASVs | 5,31 | 6,32 | 12,77 | 21,53 |
| n. of unassigned ASVs | 2450 | 1522 | 923 | 5 |
| unassigned ASVs % of ASVs | 4,68 | 4,14 | 7,09 | 0,19 |
| % of ASVs assigned to Domain | 100,00 | 100,00 | 100,00 | 100,00 |
| % of ASVs assigned to Phylum | 94,68 | 95,04 | 92,64 | 99,77 |
| % of ASVs assigned to Class | 94,07 | 94,31 | 92,32 | 99,50 |
| % of ASVs assigned to Order | 91,55 | 91,27 | 90,88 | 98,91 |
| % of ASVs assigned to Family | 89,23 | 88,74 | 89,04 | 97,21 |
| % of ASVs assigned to Genus | 79,04 | 78,25 | 80,14 | 84,76 |
| % of ASVs assigned to Species | 40,88 | 42,00 | 38,68 | 36,11 |

**Fig 6. Reached rank efficiency of the taxonomic annotation across the ASV pools.** Proportions of successful taxonomic annotations for the global data and their partials (percentage of ASVs reaching a name for that rank) across the different levels of systematics. The green-to-red colour coding of the conditional formatting is used as described in the legend of Fig 2.

the bioinformatics annotations always yield a significant number of unassigned reads. Furthermore, as previously mentioned, not all variants reach the same depth of classification when a suggested lineage is obtained. In other words, the finest rank given by the output may be limited to the genus (as is most often the case), or even stop at the family, order, class, or, rarely, the phylum level. The results of this identification success for the present project are shown in Fig 6. The ASVs of the project are separated into those specific to insects or habitats, and those shared by both. Some interesting differences arose from this separation.

The shared variants showed a much higher proportion of named taxa and a much better success rate in achieving deep rank assignment. This suggests that bacteria associated with both insects and the environment are more accurately represented in current databases than the general average. Additionally, the percentage of different taxonomic names for bacterial ASVs specific to insects (12.77%) is higher than for general ASVs. The fact that more microbiome studies are currently being conducted than bacterial reports on mountain or glacier habitats may have an impact on how such results are interpreted.

## Overall considerations on the tested hypotheses

One of the main objectives of the present study was to assess whether the gut microbes of chironomids reflect environmental communities or form distinct assemblages associated with the host. The alternative scenarios of symmetry (mirroring the environment) vs. specificity (distinct insect-associated assemblages) suggest the latter, as supported by the data. These indicate a very high degree of community divergence and, consequently a rather limited overlap.

With regard to the 'who eats whom' question, evidence has been found to rule out the hypothesis that the bacterial checklist in the larvae's gut would correspond to their diet. The minimal overlap in sequence variants between the environment and larvae (Table 3, Fig 4) renders this assumption unlikely. Therefore, as with humans and other animals in less extreme and more interactive habitats, the observed taxa can be considered a bona fide set of species involved in the host's health-related physiological functions. The chironomid microbiome, essentially featured by the larval digestive tract, would therefore entail a role for those bacteria as 'cooks' rather than as leftovers from the meal.

It is also interesting to note that the ASV-based data, on which the proportions of the evidence for this partitioning are based, can be considered more reliable than any details of nomenclatural systematics. In fact, lineage assignment is a bioinformatic inference based on the accuracy and completeness of current databases. Conversely, irrespective of their

possible taxonomic identities, the sequence variants (ASVs) represent the true underlying data, as they stem from the sequencing run output of the experiment itself and consist of primary physical data in the form of nucleotide sequences.

In order to better address the reasons why the 'we are what we eat' tenet would not necessarily hold when using DNA to determine diet, some considerations must be made. Previously, we demonstrated that analyzing the faeces of cattle grazing in alpine meadows and targeting the plant chloroplast trnL intron region could faithfully reconstruct the floral composition of the ingested vegetation and its seasonal shifts [49]. Other authors have demonstrated that DNA-based analyses can reveal bryophytes grazed by invertebrates in glacier habitats [82]. Regarding the retrieval of diet-borne bacterial DNA, our work with fattening quails involved comparing the birds' faeces to the initial diet. We were able to identify the main bacteria supplied by the diet in the quail excreta, alongside a large number of others belonging to the resident microbiome [83]. However, in the case of insect and microbe trophic intersections, our experience was with an environment that shares characteristics with the present investigation. The common aspect is inferring the food source of insects living in stressful, low-productivity habitats. In that study, we dealt with an endemic hygropetric beetle (*Cansiliella servadeii*) that lives in karstic caves and feeds on a submerged carbonate speleothem called 'moonmilk'. Despite moonmilk being endowed with a complex microbial mat community and apparently being the only food source for that insect, which spends most of its time grazing over it, we found practically no taxonomic overlap between the insect's gut bacteria and those colonizing the moonmilk [84]. These different pieces of evidence suggest that while DNA extracted from faeces or gut contents can reveal an animal's dietary composition, as was the case with bovines and quails, this only seems to apply to animals that ingest bulky food in large quantities. In contrast, the feeding of aquatic insects in low-productivity environments such as caves and glaciers appears to rely mostly on a detritus/dissolved organic matter food web, in which decaying cell debris rather than intact DNA-loaded bacterial cells represents the actual intake for invertebrates. This suggests that, although these larvae could subsist on partly degraded microbial remnant material, their gut microbiome might not be a suitable proxy for determining their diet when matched with the originally ingested DNA.

These results are consistent with a survey of many marine invertebrates, which showed that their bacteria differ from those in the surrounding environment. The authors also found that coexisting invertebrates, even when not related to each other, tend to share many of the same bacteria. This suggests that communities of microorganisms that are associated with animals but not linked to any specific host lineage are the main drivers of these ecological interactions. Co-occurring invertebrates were found to share a much higher proportion of bacterial species with each other than with their environment. Host identity would therefore be a very minor factor in shaping these communities, which, unlike large animals, do not exhibit phylosymbiosis [85].

The finding of a highly specific, low-diversity chironomid microbiome, which is strongly uncoupled from the diverse external community (only 4.9% shared ASVs, with insect communities having sevenfold lower diversity), suggests a critical host-driven selection process. A stable, specialized core community has significant ecological implications for the host's fitness in the face of glacial retreat and environmental change. These involve: (a) resilience to glacial melt; (b) colonization of new environments. A table, outlining the mechanisms by which microbes can be of support in these actions and the ensuing consideration is provided in the Supporting Information Table S1 in S1 File.

As regards the possible ecological roles of the observed taxa, these remain speculative because functional roles derived from 16S rRNA gene data are inferences, and not confirmed activities. However, the consistent and unique enrichment of specific bacterial taxa within the insect gut (constituting almost 25% of strictly insect-specific taxa) supports their functional necessity. Their known metabolic capabilities, when linked to the host's ecological niche (detritivory in oligotrophic, cold environments), provide circumstantial evoidences for their roles. The dominant, recurring members of the insect core group are *Providencia*, *Serratia*, *Massilia*, and *Flavobacterium*. Their likely exerted ecological roles have been listed and commented in Supporting Information Tab. S2 in S1 File. Conversely, for the corresponding inference regarding the taxa that dominate the surrounding habitat, parallele considerations are outlined in Supporting Information tab. s3. in S1 File.

## Conclusions

In essence, the present investigation allows the following remarks to be traced from several points of evidence. The glacier environment is surprisingly rich in microbial diversity, comparable to levels observed in far more productive ecosystems. There is relatively little overlap between its bacterial community and the much less diverse chironomid-associated ones, and specificity prevails over symmetry.

There were few differences between insect species or between communities at different sites. There was no evidence regarding the diet of chironomids in relation to the microbiota of the habitat. This suggests that the food resource for these chironomids probably consists mostly of dissolved organic matter and detritus of various origins, rather than intact bacteria with sequenceable genomes, as is the case with boluses from plants or other food for ruminant herbivores or birds.

The implications of the present results for glacier ecology and chironomid biology are that these symbiotic bacteria may protect these insects from environmental stressors, even of natural origin (e.g. extremely low temperatures), by promoting the digestion of food with low nutritional value and/or contributing to the insect's health and fitness through participation in its metabolism. Knowing more about the array of hosted microbiota could support studies devoted to the transmission of potentially critical microbial taxa from adult to egg to hatched offspring, providing a more advanced understanding of their mutual relevance in fragile and endangered ecosystems.

## Supporting information

**S1 File. Supporting information Lencioni et al.**
(DOCX)

## Acknowledgments

Alessandra Franceschini is gratefully acknowledged for technical support and logistic assistance during the expedition.

## Author contributions

**Conceptualization:** Valeria Lencioni, Andrea Squartini.

**Data curation:** Isabel Martinez-Sañudo, Luca Mazzon, Andrea Squartini.

**Formal analysis:** Alessandra Tondello, Isabel Martinez-Sañudo, Saptarathi Deb, Lucia Giagnoni, Luca Mazzon.

**Investigation:** Valeria Lencioni, Alessandra Tondello, Isabel Martinez-Sañudo, Saptarathi Deb, Lucia Giagnoni.

**Methodology:** Valeria Lencioni, Isabel Martinez-Sañudo, Saptarathi Deb, Luca Mazzon, Andrea Squartini.

**Project administration:** Augusto Zanella.

**Resources:** Augusto Zanella, Giuseppe Concheri, Piergiorgio Stevanato, Luca Mazzon, Andrea Squartini.

**Software:** Saptarathi Deb.

**Supervision:** Valeria Lencioni, Luca Mazzon, Andrea Squartini.

**Validation:** Valeria Lencioni, Alessandra Tondello, Luca Mazzon, Andrea Squartini.

**Visualization:** Valeria Lencioni, Andrea Squartini.

**Writing – original draft:** Valeria Lencioni, Andrea Squartini.

**Writing – review & editing:** Valeria Lencioni, Alessandra Tondello, Isabel Martinez-Sañudo, Saptarathi Deb, Lucia Giagnoni, Augusto Zanella, Giuseppe Concheri, Piergiorgio Stevanato, Luca Mazzon, Andrea Squartini.

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
