## [Decision Letter · Decision Letter 0]

7 Jul 2025

We look forward to receiving your revised manuscript.

Kind regards,

Luigimaria Borruso

Academic Editor

PLOS ONE

Journal Requirements:

[Grant BioMiti 2018, issued from Adamello-Brenta Natural Park, Italy, awarded to A.Z.]. 

[This work was made possible in part by funding from the Adamello-Brenta Natural Park, Project Grant BioMiti 2018. Alessandra Franceschini is gratefully acknowledged for technical support and logistic assistance during the expedition.]

[Grant BioMiti 2018, issued from Adamello-Brenta Natural Park, Italy, awarded to A.Z.]. 

4. We note that Figure 1 in your submission contains map/satellite images which may be copyrighted. All PLOS content is published under the Creative Commons Attribution License (CC BY 4.0), which means that the manuscript, images, and Supporting Information files will be freely available online, and any third party is permitted to access, download, copy, distribute, and use these materials in any way, even commercially, with proper attribution. For these reasons, we cannot publish previously copyrighted maps or satellite images created using proprietary data, such as Google software (Google Maps, Street View, and Earth). For more information, see our copyright guidelines: http://journals.plos.org/plosone/s/licenses-and-copyright.

Reviewers' comments:

Reviewer's Responses to Questions

**Comments to the Author**

1. Is the manuscript technically sound, and do the data support the conclusions?

Reviewer #1: No

Reviewer #2: Yes

2. Has the statistical analysis been performed appropriately and rigorously?

Reviewer #1: No

Reviewer #2: I Don't Know

3. Have the authors made all data underlying the findings in their manuscript fully available?

Reviewer #1: Yes

Reviewer #2: Yes

4. Is the manuscript presented in an intelligible fashion and written in standard English?

Reviewer #1: No

Reviewer #2: Yes

Reviewer #1: High microbial diversity in glacial habitats uncoupled from the specialized microbiomes of resident chironomid fauna

The manuscript presents an interesting investigation into the microbial ecology of glacial systems, focusing on the comparison between environmental bacterial communities and those associated with chironomids inhabiting glacial environments. The research addresses an important and little-explored topic, particularly with regard to the possible relationships between host-associated microbiomes and those present in the surrounding extreme environment. The topic is certainly of interest to a wide scientific audience, particularly in the fields of microbial ecology and environmental biology. However, the manuscript requires substantial revisions to improve clarity, text organisation and consistency between the various sections, as well as to justify certain methodological choices and interpretations more effectively. A revision of the English is essential. More scientific and direct English is needed, with simple and more schematic sentences. Along the entire manuscript, all sentences can be made simpler and clearer. Once the structure and content have been finalised, I recommend a review by a native English speaker.

However, a major gap in this work is the lack of statistical data to support the findings and the lack of images that clearly represent the objectives of the work (e.g., NMDS for community overlap between hosts and environment, Venn diagrams to determine how many ASVs, species or genera are shared, etc.). These aspects need to be integrated.

Below are detailed comments by section and specific suggestions line by line to help the authors strengthen their work.

Introduction

The introduction is grammatically correct, with a few minor exceptions listed below. However, I find the writing style overly complex. The language is not very scientific; it seems more ‘poetic’ and, in a context where simple and direct concepts need to be expressed for a good understanding of the scientific context, it is distracting and confusing. There are also very long sentences that slow down readability, or sentences that seem to have been translated word for word. Here are some examples:

1) Line 54: ‘In spite of the concept of low temperatures as keepers of life stillness...’. It does not clearly explain the concept of low temperatures blocking biological activity. Why not write something like: ‘Despite the common perception of low temperatures as preserving biological inactivity...’? This is just one example. Another is line 72: ‘As a destiny of deglaciated foreland...’. It would be better to write more directly.

2) Line 60: ‘Meta-analyses have evidenced sets of functions...’. ‘Evidenced’ seems to have been translated directly. It would be better to use ‘identified’, ‘highlighted’, ‘showed’, etc. Line 70: ‘An inspection of the prokaryotic benthic diversity...’. Use ‘analysis’ or ‘assessment’ instead of ‘inspection’.

3) Lines 79-83 express a simple concept but are written in an overly complex manner and require multiple readings. They could be simplified with ‘Glacial environments also host a variety of invertebrate animals, which interact closely with microbial communities. In these animals, microbes are primarily found in the gut and may be either resident, contributing to physiological or symbiotic functions, or transient, originating from ingested food.’ There are also unnecessary additions, such as ‘as in any other context’ (line 80), and non-scientific definitions, such as ‘The glacial setting’ (line 79). Glaciers are not settings but environments. Another strange construction: ‘communicative interplay between the two dimensions of the outer environment and the inner microbiome’ (lines 80-81). Another example lines 108-110: “Chironomids from cold habitats have also been studied in relation to their feeding using carbon-stable isotopes. In this respect, the past glacier dynamics were found to be the main driver of autochthonous primary productivity”. This could be replaced with ‘Carbon stable isotope studies have shown that feeding patterns in chironomids are influenced by past glacier dynamics, which shape autochthonous primary productivity.’

The text also contains sentences such as ‘Covering almost 10% of the earth's surface, glaciers offer a valuable ground for comparative research, from which meta-analyses have evidenced sets of functions with significant recurrence, among which nitrogen fixation, aerobic nitrite oxidation and nitrification’ (lines 59-61) and ‘An inspection of the prokaryotic benthic diversity comparing glacial and non-glacial streams indicated a higher richness for the latter’ (lines 70-71), which are not expanded upon in the following sections and do not seem to serve much purpose in the introduction. In the first case, the functions mentioned are not reconsidered for further investigation. In the second case, non-glacial areas have never been mentioned before, so why mention this comparison?

Finally, I think that changing the order of the topics in the introduction could enhance the work. The focus is on chironomids and how their microbiota compares with that of the external environment. So, I would start with chironomids and their presence in glacial environments, and why they are better subjects than other organisms for this type of study -> intestinal microbiota -> glacial environments and environment-insect links in these extreme environments. This is just a suggestion, but generally, the introduction needs to improve its flow and establish much stronger and more direct connections between the cited topics.

56 melted water -> melted ice

60 from which, meta-analyses -> remove the comma

62 phyla as -> phyla such as

63 genera as -> genera such asù

65 Ferruginibacter genus have been also signaled -> Ferruginibacter genus have been also reported? Observed?

88-89 even the often neglected dimension of individual variation, is the source… -> even the often-neglected individual variation is the (remove dimension of and comma)

99 offsprings -> offspring

113 at characterizing -> to characterize

Materials and Methods

127-132 Is there any additional information on the sites for a more detailed description? Average temperatures and precipitation, exposure, any data that can be used to describe the sites and assess whether they are similar or different, and to contextualise the results obtained.

132 Remove comma between Agola coordinates and Fig citation.

148-151 wherever necessary, what does it mean? It is better to split the sentence and explain better, like: “Chironomid larvae were collected using a 100-µm mesh pond net. Taxonomic identification was initially done via a morphological approach (microscopic examination of larval features). When morphology was not enough for an appropriate identification, we performed molecular identification by extracting the larval DNA and amplifying the mitochondrial cytochrome oxidase I (COI) gene, following the protocol of [25]”. Add somewhere the number of collected samples: e.g. 12 environmental samples, 34 larvae and 2 adults.

157 Remove the comma at the end of the subtitle.

151 and 159 The methods are too vague. Describe at least the kit used for the extraction and the primers.

159-164 One question: Did you include negative controls during DNA extraction (both for insects and the environment) to check for contamination? Low-biomass samples, such as glacier ice, can be prone to contamination from reagents, so it is common to perform blank extractions. If you did, please indicate how they were handled (and presumably no amplification in them). If you did not include them, this limitation could be worth acknowledging later. I recommend at least noting whether a blank was performed during the sequencing workflow and what the result was (hopefully, negligible reads). You may consider also this relevant paper that emphasizes the importance of including and sequencing negative controls in studies involving low-biomass alpine environments: https://doi.org/10.1016/j.ecolind.2022.109737,
https://doi.org/10.1016/j.scitotenv.2023.168159.

170 Remove comma after sets.

168-170 Explain the methodology better. Was the PCR program the same for both primer pairs? The two primer pairs were used together in a multiplex, or have you run two PCRs per sample? Were the pooled amplicons from the amplification of the two different primer pairs? Have you run PCR replicates? They are important for low biomass samples like glacier water or small insect guts.

175 Xpress Plu 9… and Express Bacorde Kit. Which is correct? Xpress or Express?

183 Expected reads length?

192 “the feature and taxonomy tables were exported” is not necessary.

200 Which model do you use on the SHAMAN online utility?

201 Sequences have been deposited in NCBI’s Sequence Read Archive under accession PRJNA1222183.

I would suggest adding statistical analyses to strengthen the interpretation of differences between microbial communities associated with insects and the environment. It would be appropriate to include a Non-metric Multidimensional Scaling (NMDS) analysis based on Bray–Curtis distances. This analysis would have provided an effective visual representation of community structure, highlighting any separation between groups. In addition, applying a statistical test such as PERMANOVA (adonis in R) would have allowed a quantitative assessment of whether the differences observed between chironomid species and sites were statistically significant. These approaches are standard in microbial ecology analyses and would have added statistical rigour and robustness to the conclusions. Furthermore, including a Venn diagram and similarity indices (e.g., Jaccard) between groups would have further supported the observations on the low community overlap.

In any case, statistical analyses are necessary and fundamental for a correct interpretation of the differences.

Results

The results are interesting, but I regret to say that this section needs to be completely revised. This part of the manuscript mixes results, discussion and even materials and methods (see lines 378-383, where it emerges that another analysis was performed). The results section must follow closely the materials and methods section. If the first paragraph of the materials and methods section discusses extraction and amplification, the results section must discuss what was obtained in terms of the quality of the results (e.g., all samples were extracted and amplified, or some samples yielded too low and were discarded), numerical data, and comparisons supported by statistical data. And so on, following the structure of the materials and methods. In addition, many descriptive parts should go in the discussion and not in the results. Furthermore, the sentences are complex and not very scientific. Here, you need to present the raw data, almost like a list.

284 Since you have Table 2, Figure 4 is not necessary. The table is more visually appealing.

Discussion

The discussion should be better organized following the main hypotheses. The discussion is weak and does not properly explore the results.

Eg. 460-474 Interesting but out of the objective of the manuscript.

476 This is not in contrast with… Try to be more direct in your sentences. The way they are structured now, they are convoluted and require multiple readings to be understood.

Eg. 486-490 The examples (quails, forests, air) are not directly relevant to insects or glacial environments. The paragraph attempts to justify that the host species and site did not influence the microbiome, but it does so by referring to very different systems with different biological contexts. It does not clearly link back to the main objective of the study: whether the gut microbes of chironomids reflect environmental communities or form distinct assemblages associated with the host.

The main objectives of the paper are taken into consideration in the discussion, but they are treated marginally and get lost in digressions that are not strictly related to the objective of the work. Furthermore, works that have nothing to do with the context are often cited.

Be sure to explicitly circle back to the “symmetry vs specificity” hypothesis. State clearly which scenario your results supported. From the data, it’s clear that specificity prevailed (very high divergence, low overlap). Perhaps mention what is known about insect gut microbiomes in harsh environments or in detritivores. It’s known that many detritivorous insects rely on a limited set of gut symbionts (or sometimes none obligate). If chironomids harbour fewer ASVs, this fits the idea that many insects have relatively simple microbiomes (line 244-248 you stated that). You can cite general insect microbiome literature to say this finding of low diversity gut vs outside is consistent with insects filtering their microbiota or maintaining only certain bacteria. If any specific genera dominated chironomid guts in your data, discuss them.

The absence of differences between chironomid species or sites is a noteworthy finding, as one might expect different species (especially those with different feeding habits) or different glacial sites (with different conditions) to harbour different microbiomes. The fact that this is not the case suggests a strong commonality. Discuss possible reasons:

- Could it be that all these chironomid species feed on very similar resources (e.g., microbial mats, detritus) in the glacier streams, thus acquiring a similar microbiome?

- Or perhaps the host filters the microbiome so strongly that regardless of environment, only certain bacteria can thrive in the gut (due to physiological conditions like anoxia, antimicrobial enzymes, etc.).

- It might also be partially methodological: if each species had few samples, maybe differences weren’t detectable – if so, mention that sample size could be a limiting factor (but don’t undermine your result too much; just a cautious note if appropriate).

- If any species were aquatic vs semi-terrestrial, could that matter? (Likely all larvae were aquatic, though.)

Cite implications and future directions.

Figure 1: Scale bar and north indication are missing. I would remove the black background.

Figure 2: Explain all colours in the caption. Word colours and taxa and abund gradient.

Figure 3: Chao1 axes name bracket is missing. Statistic is missing.

Figure 4: Not necessary, the table in the text looks better.

Reviewer #2: Dear Authors

Thank you for the effort you have put in this paper. This manuscript provides original insights into the ecological separation between glacier microbial communities and insect microbiomes, which are highly relevant in the context of global warming and glacial retreat.

I have some concern, as it needs some improvements:

*Enrich the introduction or discussion with Important biological and morphological aspects specific to the species in question. This may facilitate the interpretation of the findings.

*Use Well-drawn speculative interpretations to avoid firm conclusions.

*better statistical support for key results, especially for beta diversity (e.g., PERMANOVA9.

*Better visualization for beta diversity (NMDS maybe).

*Improve clarity by shortening some sentences or dividing them into smaller parts (See Below)

*the introduction is Simple and well-written, both cohesion and coherence.

*Line 26–28: avoid “to understand whether bacteria … could be considered food” (this phrasing implies a strong dietary hypothesis that the data don’t fully support).

*Lines 45–50: Avoid conclusions like “data showed no clues,” as this will decrease the value of your work hypothesis. Consider using “suggests” or “provides limited support for…”

*Lines 53–59: shorten the sentences or divide them into shorter sentences to improve readability.

*Lines 72–78: Avoid repetition and be more direct. This passage can be shortened to fewer lines

*Line 105–112: Try here to answer the question, why does this study matter? Why are you hypothesizing a confrontation between the Chironomidae gut and the Environmental Microbiome? Why did you expect that the Chironomida diet could be directly from bacteria in its habitat?

*Line 121–124: poor description of the hypothesis, and it would be better to be placed before the research question/objectives

*More detailed information on the sampling protocol is needed to ensure reproducibility

*Good coverage of multiple glacier systems.

*Line 129: report how many larvae here also.

*Line 151: Why were adults included, and also why did you base our work on larvae?

*Line 192: and beta diversity ??

*Line 230–232: Clusterings are described, but some statistical evidence and significance (e.g., p-values or clustering robustness, PERMANOVA).

*Line 239-242: Again, what was the purpose of using adults?

*I would like to see here a PCA or NMDS with a PERMANOVA plot for beta diversity. It would be more representative and clearer than a dendrogram

Lines 316–321, 394–396, do you think that rapid digestion or DNA fast degradation may have a role in the observed result? (For the discussion)

Lines 467–471: The discussion of atmospheric deposition as a driver of high bacterial richness in glacial water is speculative and not supported by any direct evidence in this study. It should be introduced more as an explanation.

Lines 490–495: Avoid making definitive conclusions. Rephrase to acknowledge limitations of 16S metabarcoding in identifying food-derived taxa.

In conclusion, is the similarity of 4-5 % a good indicator of similarity or not? How much is good similarity? Does similarity mean direct feeding behavior and diet preference, or is it a coincidence or even indirect?

For that, in the introduction, it would be helpful if you add some morphological aspects of Chironomidae, such as feeding behaviour and mouth parts for both larvae and adults. From where you start your hypothesis? are we assuming that the Chironomid feeds on soil? OM? Slure?..rthis aspetc must be discussed

Is this a sign of feeding behavior, perhaps they like, or choose voluntarily which microbiome to feed on?

Fig3. I would use the same order of groups in B as in A (insect, pond, and finally, glacier).

Line 361- and 491-: table 4 Did you check if any of these unique Chironomid gut microbiome taxa are associated with environmental studies in the area or nearby in other studies?

Or

*Did you check at least their ecological groups? Can they be free in the environment, or must they have a host? in which environments they drive better?

*For example, it is surprising that Janthinobacterium, which are commonly found in glaciers and lakes, are not present in the environmental samples here, but instead in the gut.

*Did you check for any seasonal impact?

*When did you perform the sampling?

*Other genera are common in thermophilic environments, thus did you check the life cycle of the Chironomid?

*Are they migratory, or long-distance travelers?

*How long do adults live? How long does it take from adult hatching to laying eggs?

*They are present in glaciers, but is it their preferred habitat?

*Line 459: For more academic tone, change “In the last few years, it has become clear…” to “Recent studies have increasingly shown that…”

*Line 515–519: The moonmilk beetle comparison is interesting but need more clarification, is it relevance to glacier ecosystems or insect ecology.

*Line 498: “we definitely conclude” is too strong, replace with “we propose” or “we interpret”.

*Lines 527–530: The sentence structure is long and dense. Consider breaking it.

The closure of the discussion misses:

*What does this mean for glacier ecology or chironomid biology

*How could these findings be extended (e.g., functional profiling, manipulation of microbiomes, longitudinal studies)

**Do you want your identity to be public for this peer review?** For information about this choice, including consent withdrawal, please see our Privacy Policy

Reviewer #1: No

Reviewer #2: No

---

## [Author Response · Author response to Decision Letter 1]

16 Sep 2025

PONE-D-25-16248

Journal Requirements:

ANSWER: The text format has been uniformed to the required templates.

ANSWER: The following information has been added: “The permit to sample from the site has been released by Dr. Cristiano Trotter, the Director of Parco Naturale Adamello Brenta, Via Nazionale, 24, 38080 Strembo (TN).”

[Grant BioMiti 2018, issued from Adamello-Brenta Natural Park, Italy, awarded to A.Z.].

ANSWER: The following statement has been added: "The funders had no role in study design, data collection and analysis, decision to publish, or preparation of the manuscript."

The amended role of the funder has been stated in the cover letter.

[This work was made possible in part by funding from the Adamello-Brenta Natural Park, Project Grant BioMiti 2018. Alessandra Franceschini is gratefully acknowledged for technical support and logistic assistance during the expedition.]

[Grant BioMiti 2018, issued from Adamello-Brenta Natural Park, Italy, awarded to A.Z.].

ANSWER: We have removed the funding-related text from the Acknowledgments as requested, and indicated it in the cover letter.

4. We note that Figure 1 in your submission contains map/satellite images which may be copyrighted. All PLOS content is published under the Creative Commons Attribution License (CC BY 4.0), which means that the manuscript, images, and Supporting Information files will be freely available online, and any third party is permitted to access, download, copy, distribute, and use these materials in any way, even commercially, with proper attribution. For these reasons, we cannot publish previously copyrighted maps or satellite images created using proprietary data, such as Google software (Google Maps, Street View, and Earth). For more information, see our copyright guidelines: http://journals.plos.org/plosone/s/licenses-and-copyright.

ANSWER: The mentioned image in Fig. 1 has been substituted with a non-satellite physical map.

Reviewer #1:

The manuscript presents an interesting investigation into the microbial ecology of glacial systems, focusing on the comparison between environmental bacterial communities and those associated with chironomids inhabiting glacial environments. The research addresses an important and little-explored topic, particularly with regard to the possible relationships between host-associated microbiomes and those present in the surrounding extreme environment. The topic is certainly of interest to a wide scientific audience, particularly in the fields of microbial ecology and environmental biology.

ANSWER: We acknowledge these appreciative comments

However, the manuscript requires substantial revisions to improve clarity, text organisation and consistency between the various sections, as well as to justify certain methodological choices and interpretations more effectively. A revision of the English is essential. More scientific and direct English is needed, with simple and more schematic sentences. Along the entire manuscript, all sentences can be made simpler and clearer. Once the structure and content have been finalised, I recommend a review by a native English speaker.

ANSWER: The structure of the manuscript, along with its language and style aspects, has been thoroughly revised. The version showing the highlighted changes in yellow will be useful to appreciate the capillary editing that has been applied throughout the paper.

However, a major gap in this work is the lack of statistical data to support the findings and the lack of images that clearly represent the objectives of the work (e.g., NMDS for community overlap between hosts and environment, Venn diagrams to determine how many ASVs, species or genera are shared, etc.). These aspects need to be integrated.

ANSWER: As will be specifically highlighted below, the statistical analyses (NMDS ordination plot, PERMANOVA, ANOSIM, PERMDISP) have been performed, and the results, showing the significant p values for the separation of each sample category, have been added in the new fig. 4. A Venn diagram (Fig. S3, Supplementary information) and Jaccard distances (text) have been added as well.

Below are detailed comments by section and specific suggestions line by line to help the authors strengthen their work.

Introduction

The introduction is grammatically correct, with a few minor exceptions listed below. However, I find the writing style overly complex. The language is not very scientific; it seems more ‘poetic’ and, in a context where simple and direct concepts need to be expressed for a good understanding of the scientific context, it is distracting and confusing. There are also very long sentences that slow down readability, or sentences that seem to have been translated word for word. Here are some examples:

1) Line 54: ‘In spite of the concept of low temperatures as keepers of life stillness...’. It does not clearly explain the concept of low temperatures blocking biological activity. Why not write something like: ‘Despite the common perception of low temperatures as preserving biological inactivity...’? This is just one example. Another is line 72: ‘As a destiny of deglaciated foreland...’. It would be better to write more directly.

ANSWER: The suggestions are sound, and sentences have been accordingly revised.

2) Line 60: ‘Meta-analyses have evidenced sets of functions...’. ‘Evidenced’ seems to have been translated directly. It would be better to use ‘identified’, ‘highlighted’, ‘showed’, etc. Line 70: ‘An inspection of the prokaryotic benthic diversity...’. Use ‘analysis’ or ‘assessment’ instead of ‘inspection’.

ANSWER: Correction made (highlighted). The second sentence is no longer present as suggested in a subsequent query.

3) Lines 79-83 express a simple concept but are written in an overly complex manner and require multiple readings. They could be simplified with ‘Glacial environments also host a variety of invertebrate animals, which interact closely with microbial communities. In these animals, microbes are primarily found in the gut and may be either resident, contributing to physiological or symbiotic functions, or transient, originating from ingested food.’

ANSWER: The text has been revised as advised.

There are also unnecessary additions, such as ‘as in any other context’ (line 80), and non-scientific definitions, such as ‘The glacial setting’ (line 79). Glaciers are not settings but environments.

ANSWER: The unnecessary words as well as the ‘glacial setting’ expression, are no longer in the text as the sentence has been replaced by the one suggested in the previous query.

Another strange construction: ‘communicative interplay between the two dimensions of the outer environment and the inner microbiome’ (lines 80-81).

ANSWER: That sentence has also been removed because of the above text substitution.

Another example lines 108-110: “Chironomids from cold habitats have also been studied in relation to their feeding using carbon-stable isotopes. In this respect, the past glacier dynamics were found to be the main driver of autochthonous primary productivity”. This could be replaced with ‘Carbon stable isotope studies have shown that feeding patterns in chironomids are influenced by past glacier dynamics, which shape autochthonous primary productivity.’

ANSWER: The text has been revised as suggested.

The text also contains sentences such as ‘Covering almost 10% of the earth's surface, glaciers offer a valuable ground for comparative research, from which meta-analyses have evidenced sets of functions with significant recurrence, among which nitrogen fixation, aerobic nitrite oxidation and nitrification’ (lines 59-61) and ‘An inspection of the prokaryotic benthic diversity comparing glacial and non-glacial streams indicated a higher richness for the latter’ (lines 70-71), which are not expanded upon in the following sections and do not seem to serve much purpose in the introduction. In the first case, the functions mentioned are not reconsidered for further investigation. In the second case, non-glacial areas have never been mentioned before, so why mention this comparison?

ANSWER: We agree, and both sentences have been removed.

Finally, I think that changing the order of the topics in the introduction could enhance the work. The focus is on chironomids and how their microbiota compares with that of the external environment. So, I would start with chironomids and their presence in glacial environments, and why they are better subjects than other organisms for this type of study -> intestinal microbiota -> glacial environments and environment-insect links in these extreme environments. This is just a suggestion, but generally, the introduction needs to improve its flow and establish much stronger and more direct connections between the cited topics.

ANSWER: The order of topics has been restructured as suggested.

56 melted water -> melted ice

ANSWER: Correction made.

60 from which, meta-analyses -> remove the comma

ANSWER: That sentence has been removed in the restructured introduction.

62 phyla as -> phyla such as

ANSWER: Correction made.

63 genera as -> genera such as

ANSWER: Correction made.

65 Ferruginibacter genus have been also signaled -> Ferruginibacter genus have been also reported? Observed?

ANSWER: Correction made.

88-89 even the often neglected dimension of individual variation, is the source… -> even the often-neglected individual variation is the (remove dimension of and comma)

ANSWER: Correction made.

99 offsprings -> offspring

ANSWER: Correction made.

113 at characterizing -> to characterize

ANSWER: Correction made.

Materials and Methods

127-132 Is there any additional information on the sites for a more detailed description? Average temperatures and precipitation, exposure, any data that can be used to describe the sites and assess whether they are similar or different, and to contextualise the results obtained.

ANSWER: A table (new Table 1, shown below) has been added reporting all the available information. Regarding meteorological data, there are no permanent meteorological stations around the three sites.

Tab. 1. Physico-chemical properties of the sampling sites. The data refer to the sampling dates presented in Tab. 2, with the exception of the water temperature (Mean Tw_max) whose value refers to the mean of the 3 weeks before the sampling, collected by dataloggers. Q: water discharge (absent in Agola being it a static pond and not a flowing kryal stream).

Sampling site Agola (pond) Mandrone (kryal) Amola (kryal)

Altitude site (m asl) 2596 2569 2540

Distance from the source (km) 0.02 0.05 0.14

Suspended sediments (mg/l) 2.60 76.80 131.00

Mean Tw_max (°C) 4.5 ± 1.5 1.2 ± 0.5 1.6 ± 0.4

Chlorophyll a (µg/cm2) 0.09 0.48 0.33

pH (20°C) 8.20 6.9 6.6

Conductivity (μS/cm) 72.0 5.5 13.4

SiO2 (mg/l) 0.3 1.1 1.2

Q (m3/s) - 3.8 1.1

132 Remove comma between Agola coordinates and Fig citation.

ANSWER: Correction made.

148-151 wherever necessary, what does it mean? It is better to split the sentence and explain better, like: “Chironomid larvae were collected using a 100-µm mesh pond net. Taxonomic identification was initially done via a morphological approach (microscopic examination of larval features). When morphology was not enough for an appropriate identification, we performed molecular identification by extracting the larval DNA and amplifying the mitochondrial cytochrome oxidase I (COI) gene, following the protocol of [25]”. Add somewhere the number of collected samples: e.g. 12 environmental samples, 34 larvae and 2 adults.

ANSWER: The text has been revised as suggested.

157 Remove the comma at the end of the subtitle.

ANSWER: Correction made.

151 and 159 The methods are too vague. Describe at least the kit used for the extraction and the primers.

ANSWER: The following text has been added: “Briefly, genomic DNA was extracted from the larval body using a QIAGEN DNAeasy Blood & Tissue Kit, following a slightly modified protocol for the purification of total DNA from animal tissue. In particular, 10 μl of a 1 M DTT (dithiothreitol) solution was added to enhance the removal of chitin, after which the samples were incubated overnight at 56 °C for the initial lysis step. The final elution was performed in 50 μl of AE buffer to increase the final DNA concentration. The mitochondrial (COI gene were amplified and sequenced using the primers LCO1490 GGTCAACAAATCATAAAGATATTGG and HCO2198 TAACTTCAGGGTGACCAAAAAATCA”.

159-164 One question: Did you include negative controls during DNA extraction (both for insects and the environment) to check for contamination? Low-biomass samples, such as glacier ice, can be prone to contamination from reagents, so it is common to perform blank extractions. If you did, please indicate how they were handled (and presumably no amplification in them). If you did not include them, this limitation could be worth acknowledging later. I recommend at least noting whether a blank was performed during the sequencing workflow and what the result was (hopefully, negligible reads). You may consider also this relevant paper that emphasizes the importance of including and sequencing negative controls in studies involving low-biomass alpine environments: https://doi.org/10.1016/j.ecolind.2022.109737,
https://doi.org/10.1016/j.scitotenv.2023.168159.

ANSWER: Yes, in this type of studies we routinely perform negative controls using the same reagents (extraction without sample and eluate sequencing), and we also did those as custom, for the batches used in the series that includes the present experiments. Such tests consistently yield numbers of reads that are below the detection level as they do not pass the quality filtering bioinformatics cutoff.

We thank you for signalling the mentioned reference, which we have added and cited about the background minimization.

170 Remove comma after sets.

ANSWER: Correction made.

168-170 Explain the methodology better. Was the PCR program the same for both primer pairs? The two primer pairs were

---

## [Decision Letter · Decision Letter 1]

3 Nov 2025

Dear Dr.  Squartini,

Thank you for submitting your manuscript to PLOS ONE. After careful consideration, we feel that it has merit but does not fully meet PLOS ONE’s publication criteria as it currently stands. Therefore, we invite you to submit a revised version of the manuscript that addresses the points raised during the review process.

We look forward to receiving your revised manuscript.

Kind regards,

Luigimaria Borruso

Academic Editor

PLOS ONE

Journal Requirements:

Reviewer's Responses to Questions

**Comments to the Author**

Reviewer #1: All comments have been addressed

Reviewer #2: All comments have been addressed

2. Is the manuscript technically sound, and do the data support the conclusions?

Reviewer #1: Yes

Reviewer #2: Partly

3. Has the statistical analysis been performed appropriately and rigorously?

Reviewer #1: Yes

Reviewer #2: Yes

4. Have the authors made all data underlying the findings in their manuscript fully available?

Reviewer #1: Yes

Reviewer #2: Yes

5. Is the manuscript presented in an intelligible fashion and written in standard English?

Reviewer #1: Yes

Reviewer #2: Yes

Reviewer #1: I thank the authors for having considered all the suggested revisions and for their substantial work on the manuscript. I just have a couple of additional recommendations.

Lines 200–206: This subsection is very short. I suggest merging it with the end of the previous one.

Figure 1: The resolution is very low. Even when downloading the high-resolution file, the two maps appear blurry. I suggest recreating it using, for example, QGIS, which allows the creation of high-resolution satellite maps.

Figure 4: I recommend keeping a white background, as in the other figures, to maintain consistency throughout the article.

Figure 5: The species names should be in italics.

Figures 5 and 6: The color description of the cells is missing in the caption, as done in Figure 2 (“The cells reporting the numbers of taxa and the read abundance are colored by the Microsoft Excel conditional formatting gradient from red (lowest numerical values) to green (highest numerical values)").

Reviewer #2: Dear Authors:

Thank you for the revised manuscript. Most of the previous comments have been addressed, and the responses were helpful. A few minor points remain:

• The text flow has improved, but some long and complex sentences remain, particularly in the discussion.

• Grammar and syntax have improved but are still somewhat dense in places.

• Line 26:28 The sentence has only been partly revised; it still implies diet. This should be fully rephrased.

• While biological interpretation has been improved, the ecological implications could be explored further for example, how microbiome specificity might affect resilience to glacial melt or colonization of new environments

• Ecological roles of taxa remain speculative.

Overall, the manuscript has improved, but addressing these remaining issues would further strengthen clarity and scientific rigor.

thank you

**Do you want your identity to be public for this peer review?** For information about this choice, including consent withdrawal, please see our Privacy Policy

Reviewer #1: No

Reviewer #2: No

---

## [Author Response · Author response to Decision Letter 2]

17 Dec 2025

Reviewer #1:

I thank the authors for having considered all the suggested revisions and for their substantial work on the manuscript. I just have a couple of additional recommendations.

Lines 200–206: This subsection is very short. I suggest merging it with the end of the previous one.

ANSWER: done

Figure 1: The resolution is very low. Even when downloading the high-resolution file, the two maps appear blurry. I suggest recreating it using, for example, QGIS, which allows the creation of high-resolution satellite maps.

ANSWER: The image resolution aspect has been fixed by re-acquiring it from the source chart. See new the new version of Fig.1.

Figure 4: I recommend keeping a white background, as in the other figures, to maintain consistency throughout the article.

ANSWER: Fig. 4 has been remade using the with background as suggested.

Figure 5: The species names should be in italics.

ANSWER: The names have been rewritten in italics.

Figures 5 and 6: The color description of the cells is missing in the caption, as done in Figure 2 (“The cells reporting the numbers of taxa and the read abundance are colored by the Microsoft Excel conditional formatting gradient from red (lowest numerical values) to green (highest numerical values)").

ANSWER: The legends have been edited to include the recommended explanations.

Reviewer #2:

Dear Authors:

Thank you for the revised manuscript. Most of the previous comments have been addressed, and the responses were helpful. A few minor points remain:

• The text flow has improved, but some long and complex sentences remain, particularly in the discussion.

• Grammar and syntax have improved but are still somewhat dense in places.

• Line 26:28 The sentence has only been partly revised; it still implies diet. This should be fully rephrased.

ANSWER: As explained in the previous round, we totally agree, but the referral to a diet function is not stated as a result but as one of the initial hypotheses (that the data ruled out as we repeatedly state throughout the paper). Honestly, we had taken into account also that hypothesis, since in such a primary production-limited habitat, speculating that aquatic larvae could graze over bacterial biofilm slime and live upon that source of carbon and energy was, in our opinion, a legitimate possibility to test.

Therefore, when , in the prior round of revision, we rewrote the mentioned sentence in the following form ”We aimed to assess the extent to which insect-borne bacteria resemble those found in icemelt water and the surrounding wet and terrestrial environments, in order to determine also whether or not bacteria in the external habitat could serve as food for these animals.” we simply expressed the truth of our initial reasoning.

Anyhow, there is no problem in modifying the sentence and avoiding any mention to a diet function and for this purpose we have changed the sentence as follows:

“We aimed to assess the extent to which insect-borne bacteria resemble those found in icemelt water and the surrounding wet and terrestrial environments, in order to determine also whether the bacteria found associated with the insects could be interpreted mainly as specific dwellers, putatively involved with active physiological functions, or also as transient cells taken in for other purposes. “

• While biological interpretation has been improved, the ecological implications could be explored further for example, how microbiome specificity might affect resilience to glacial melt or colonization of new environments

• Ecological roles of taxa remain speculative.

Overall, the manuscript has improved, but addressing these remaining issues would further strengthen clarity and scientific rigor.

ANSWER: We thank the reviewer for requesting further elaboration on the ecological implications of our findings. We have leveraged the high degree of microbial specificity we observed to provide a more definitive interpretation of the functional roles of the dominant taxa within the chironomid core microbiome and the surrounding environment.

We have added the following text and constructed three new tables, placed in the supplemementary information file:

“The finding of a highly specific, low-diversity chironomid microbiome, which is strongly uncoupled from the diverse external community (only 4.9% shared ASVs, with insect communities having sevenfold lower diversity), suggests a critical host-driven selection process. A stable, specialized core community has significant ecological implications for the host's fitness in the face of glacial retreat and environmental change. These involve: (a) resilience to glacial melt; (b) colonization of new environments. A table, outlining the mechanisms by which microbes can be of support in these actions and the ensuing consideration is provided in the Supporting Information Table S1”

TO VIEW CORRECTLY THE TABLE COLUMNS PARTITIONING IN THE NEXT LINES, PLEASE REFER TO THE ANSWERS TO REVIEWERS LETTER

Ecological concept Suggested mechanism of microbial support Specific ecological insight

Resilience to glacial melt Metabolic Stability and Detoxification: Glacial melt introduces environmental instability, including fluctuating flow rates, temperature spikes, and chemical shifts. The highly specific core microbiome acts as a physiological buffer. Taxa like Providencia and Serratia (core insect taxa) are functionally known in insect systems for their roles in detoxification and breaking down complex or inhibitory compounds, ensuring metabolic continuity even when the ingested detrital quality rapidly changes due to melt events. This stable internal community can confer resistance to environmental stress, allowing the chironomids to maintain essential physiological functions where a generalist community would fail to cope with fluctuations typical of glacier-fed streams.

Colonization of new environments Nutrient Supplementation (The 'Symbiotic Propagule'): The lack of significant difference in microbial community structure across ten chironomid species and three distant glacier sites underscores the perspective importance of the core microbiome. This core community could entail also a vital symbiotic propagule that the host has an advantage in carrying with it. By supplying essential nutrients in a consistent manner, such as nitrogen or complex carbon resources, this core could facilitate the rapid and successful colonization of newly deglaciated, oligotrophic, primary succession habitats. The host-symbiont pairing seems pre-adapted to the nutrient-scarce conditions inherent in these environments.

Tab. S1. ecological implications expected for the microbiota of the investigated envirooinment

“As regards the possible ecological roles of the observed taxa, these remain speculative because functional roles derived from 16S rRNA gene data are inferences, and not confirmed activities. However, the consistent and unique enrichment of specific bacterial taxa within the insect gut (constituting almost 25% of strictly insect-specific taxa) supports their functional necessity. Their known metabolic capabilities, when linked to the host's ecological niche (detritivory in oligotrophic, cold environments), provide circumstantial evoidences for their roles. The dominant, recurring members of the insect core group are Providencia, Serratia, Massilia, and Flavobacterium. Their likely exerted ecological roles have been listed and commented in Supporting Information tab. S2.”

Dominant core taxon Inferred ecological role in chironomid gut Insights

Providencia, Serratia (Gammaproteobacteria) Nutrient Scavenging and Detritivory: These genera are commonly associated with insect guts and are known for their high metabolic versatility, including the breakdown of complex organic compounds. In the low-quality, detritus-rich diet of the chironomid, they are likely key primary decomposers, providing the host with readily absorbable organic acids and amino acids. Their prevalence suggests a dedicated function in maximizing energy and nutrient extraction from a nutritionally poor food source, which is crucial for survival in the oligotrophic glacial stream environment.

Massilia (Betaproteobacteria) Cold-Adaptation and Immune Support: Massilia members are frequently psychrotolerant (cold-enduring) and have been implicated in various insect-microbe interactions. They likely contribute to host fitness by maintaining robust metabolic activity at the low temperatures typical of glacier-fed streams. Their role is inferred to be one of physiological support, maintaining gut homeostasis and potentially participating in defense or signaling under extremely cold conditions.

Flavobacterium (Bacteroidetes) Polymer Degradation (e.g., Chitin): Members of the Bacteroidetes phylum are renowned for their massive arsenal of glycoside hydrolases. Flavobacterium is likely selected for its capacity to break down large, tough biopolymers, such as the chitin found in fungal debris or the cell walls of microbial mats that form part of the chironomid's scraped or ingested diet. This is a highly specific enzymatic service required to release sequestered nutrients and energy that the chironomid host could not obtain otherwise, further suggesting a direct role in nutrition.

Tab. S2. Ecologically insightful roles of insect-specific core taxa

“Conversely, for the corresponding inference regarding the taxa that dominate the surrounding habitat, parallel considerations are outlined in Supporting Information Tab. S3.”

Dominant environmental taxon/group Inferred ecological role in habitat Insights

Arenimonas Primary Colonization and Nutrient Mineralization: Often found in proglacial sediments and newly deglaciated soils. This genus is a key pioneer colonizer, capable of metabolizing diverse organic compounds and mineralizing phosphorus and nitrogen, which facilitates the early stages of ecosystem development.

Methylotenera Carbon Cycling (Methylotrophy): These are specialized bacteria that utilize one-carbon compounds (e.g., methanol, methylamine). They act as critical carbon processors in the habitat, often forming syntrophic relationships that mitigate the release of methane or process specialized carbon sources available in glacial meltwater.

Nitrosomonadaceae Nitrogen Cycling: , members of this family are recognized chemoautotrophs, specifically involved in nitrite oxidation (a key step in nitrification). Their role solidifies the glacier habitat as a site of intense, externally-driven nitrogen transformation, but the lack of strong overlap with the insect gut suggests that the chironomid's core microbiome is selected to bypass reliance on this external cycle.

Vicinamibacterales Sorption and Adherence: This order often includes taxa associated with soil and sediment, likely serving a role in binding and forming the microbial mats and biofilms from which the chironomid larvae forage. Their high prevalence highlights the structural microbial component of the glacial environment, which forms the physical substrate that the chironomids scrape or ingest, but their limited presence in the gut suggests effective host selection/filtering.

Tab. S3. Ecological insights into environment-dominant taxa

Hoping to have fulfilled all the pending issues, we thank you for your editorial work and kind attention.

---

## [Editor Report · Decision Letter 2]

26 Dec 2025

High microbial diversity in glacial habitats uncoupled from the specialized microbiomes of resident chironomid fauna

PONE-D-25-16248R2

Dear Prof. Andrea Squartini

We’re pleased to inform you that your manuscript has been judged scientifically suitable for publication and will be formally accepted for publication once it meets all outstanding technical requirements.

I wish you a Happy New Year

Best Regards 

Luigimaria Borruso

Academic Editor

PLOS One

Additional Editor Comments (optional):

I strongly recommend replacing “16S rDNA” with “16S rRNA gene” throughout the whole text.

Although the sequencing is technically performed on DNA, the target of the analysis is the gene that encodes the 16S ribosomal RNA (rRNA), which constitutes the standard molecular marker for bacterial taxonomy and phylogenetic inference.

For this reason, the term “16S rRNA gene” is widely used in metabarcoding studies and has become the preferred terminology in the recent literature.
---

## [Editor Report · Acceptance letter]

PONE-D-25-16248R2

PLOS One

Dear Dr. Squartini,

I'm pleased to inform you that your manuscript has been deemed suitable for publication in PLOS One. Congratulations! Your manuscript is now being handed over to our production team.

Kind regards,

on behalf of

Dr. Luigimaria Borruso

Academic Editor

PLOS One